# [Re] Interpreting CLIP with Hierarchical Sparse Autoencoders

## Abstract

CLIP (Contrastive Language-Image Pretraining) is a multimodal model able to transform both images and text into a fixed-size vector (Radford et al., 2021). These vectors are usually uninterpretable due to polysemanticity. Sparse AutoEncoders (SAEs) enable sparser representations of vectors, while also promoting monosemanticity. The purpose of this work is to reproduce the paper *Interpreting CLIP with Hierarchical Sparse Autoencoders* (Zaigrajew et al., 2025). The authors introduce the Matryoshka Sparse Autoencoder (MSAE), which learns representations in a hierarchical fashion. In order to reproduce this work, we identify and attempt to reproduce four central claims. Our results mainly support these four claims. From this, we conclude that the results are reproducible. We also propose two extensions: (i) a model-agnostic centroid-based method to assess monosemanticity for SAE neurons, and (ii) further inspection of progressive recovery capabilities at various granularity levels.

## 1 Introduction

Traditional vision models are usually trained on images with fixed, pre-defined labels (e.g., ImageNet classes). Adding new concepts requires relabeling data and retraining, both being time-consuming tasks. Vision-language models, in particular Contrastive Language-Image Pretraining (CLIP; Radford et al., 2021) transformed this classic approach by pairing each image with a text description. This bridges the gap between vision and language through contrastive learning. By learning a multimodal embedding space (i.e., image and text), CLIP enables zero-shot generalization, allowing it to perform downstream tasks without task-specific training. However, interpretation of this CLIP embedding space remains a challenge due to dense embeddings with entangled dimensions that may entail multiple concepts (Zaigrajew et al., 2025; Goh et al., 2021; Chefer et al., 2021).

Sparse autoencoders (SAEs) offer a principled approach to addressing this interpretability challenge. They consist of an encoder that maps input $x$ to a sparse latent vector $z$ and a decoder that reconstructs $\hat{x}$ from $z$. When applied to CLIP embeddings $x$, they learn a sparse latent representation $z$ that can help identify and disentangle the main semantic features or concepts encoded by CLIP. Sparsity is enforced through several methods, such as: ReLU activations combined with $L_1$ regularization, which naturally produces zeros and penalizes all non-zero activations; and Top-K, where only the $K$ largest activations are kept while the others are zeroed.

Work by Zaigrajew et al. (2025) introduces a new architecture, namely Matryoshka SAE (MSAE). This allows models to learn coarse-to-fine hierarchical representations at multiple granularity levels—inspired by Matryoshka representation learning (Kusupati et al., 2022). This hierarchical approach addresses the limitations present in the previously mentioned SAE mechanisms. Specifically, different sparsity strategies introduce distinct shortcomings: ReLU-based SAEs exhibit activation shrinkage, whereas TopK SAEs enforce rigid sparsity constraints. Presenting a trade-off between reconstruction fidelity and sparsity, which MSAE addresses through its hierarchical design (Zaigrajew et al., 2025).

## 2 Scope of Reproducibility

From Zaigrajew et al. (2025), we examine the following claims, hereafter referred to as **C1**, **C2**, **C3** and **C4**:

- **C1: Hierarchical sparse autoencoders improve the sparsity–fidelity trade-off.** MSAE establishes a principled approach that balances reconstruction quality with sparsity in multimodal embeddings.

- **C2: MSAE learns hierarchical and progressive representations.** The model naturally captures multi-level representations, where different layers or components encode information at varying levels of abstraction.

- **C3: MSAE discovers monosemantic, human-interpretable concepts.** Neurons in MSAE correspond to coherent, interpretable semantic concepts across modalities, which can be systematically identified and validated.

- **C4: MSAE enables concept-based interventions and analysis.** The learned concepts allow controlled manipulation, similarity assessment, and systematic study of potential biases in downstream models or tasks.

## 3 Methodology

The subsequent sections 3.1, 3.2, 3.3 outline the models, datasets and hyperparameters used in this reproduction study. Section 3.4 outlines the technical setup necessary to reproduce each claim. Section 3.5 introduces the motivation and evaluation methods of our extension.

### 3.1 Model Descriptions

We reproduce all the claims stated above, using the original paper's codebase (Zaigrajew et al., 2025). The models are not re-implemented; instead, they are retrained using different configurations.

**Contrastive Language-Image Pretraining (CLIP).** Following the proposed method by Zaigrajew et al. (2025), we use pretrained CLIP ViT-L/14 pooled image and text embeddings $h_{img}, h_{text} \in \mathbb{R}^{768}$ as input representations (Radford et al., 2021). These embeddings are precomputed using fixed CLIP parameters and remain frozen throughout all experiments.

**ReLU Sparse Autoencoders (ReLU SAEs).** As baseline, the ReLU SAE, also discussed in Zaigrajew et al. (2025); Bricken et al. (2023), is used on frozen CLIP embeddings. The ReLU SAE consists of a linear encoder, followed by a ReLU activation function and a decoder. Here we denote the CLIP embeddings as $x \in \mathbb{R}^n$, then the latent activation becomes:

$$z = \text{ReLU}(W_{enc}(x - b_{pre}) + b_{enc}) \tag{1}$$

where $W_{enc} \in \mathbb{R}^{n \times d}$ denotes the encoder matrix, $b_{pre} \in \mathbb{R}^n$ the preprocessing bias, and $b_{enc} \in \mathbb{R}^d$ the encoder bias. Then the input is reconstructed by the decoder as:

$$\hat{x} = W_{dec}z + b_{pre} \tag{2}$$

where $W_{dec} \in \mathbb{R}^{d \times n}$ denotes the decoder matrix. To induce sparsity, the ReLU SAE applies $L_1$ regularization on the latent activations, where the $\lambda$ parameter controls the sparsity strength:

$$\mathcal{L}(x) := \|x - \hat{x}\|_2^2 + \lambda\|z\|_1 \tag{3}$$

**TopK Sparse Autoencoders (TopK SAEs).** Another baseline model that was used on frozen CLIP embeddings to compare the performances is TopK SAE, which induces sparsity by retraining the $K$ largest latent activations for each input and zeroing out the rest (Zaigrajew et al., 2025; Gao et al., 2025). The TopK SAE is an extension of the ReLU SAE where a TopK operation is applied on the encoder:

$$z = \text{ReLU}(\text{TopK}(W_{enc}(x - b_{pre}) + b_{enc})) \tag{4}$$

For the TopK SAE the decoder reconstructs the input similarly as for the ReLU SAE, but instead of $L_1$ regularization a basic reconstruction objective is applied:

$$\mathcal{L}(x) := \|x - \hat{x}\|_2^2 \tag{5}$$

**Matryoshka Sparse Autoencoders (MSAEs).** MSAE extends SAEs by jointly training nested sparse representations (Zaigrajew et al., 2025; Kusupati et al., 2022). We denote the CLIP dimension as $n = 768$ and use expansion factor of 8, resulting in a latent dimension $d = 6144$. Given an input representation $x \in \mathbb{R}^n$, corresponding to a frozen CLIP embedding the encoder computes a sparse latent representation:

$$z_i = \text{ReLU}(\text{TopK}_i(W_{enc}(x - b_{pre}) + b_{enc})) \tag{6}$$

where the $\text{TopK}_i(\cdot)$ returns the $i$-th largest magnitude. Meanwhile ReLU ensures latent activations are non-negative and encourages sparsity by mapping negative pre-activations to zero. The input representation is reconstructed by the decoder as:

$$\hat{x}_i = W_{dec}z_i + b_{pre} \tag{7}$$

To learn representations at multiple granularities simultaneously, MSAE computes a weighted reconstruction objective across multiple sparsity levels, where the coefficient $\alpha_i$ weights the contribution of each sparsity level to the overall loss:

$$\mathcal{L}(x) := \sum_{i=1}^{H} \alpha_i \|x - \hat{x}_i\|_2^2 \tag{8}$$

### 3.2 Datasets

A subset of the CC12M dataset is used to train all sparse autoencoders, while ImageNet-100 is used for model validation. The CelebA dataset was used for bias evaluation and the CIFAR10 dataset for testing the generalization of the progressive recovery results.

**Conceptual Captions (CC12M).** The CC12M dataset (Changpinyo et al., 2021) consists of roughly 12 million images annotated with captions. Due to broken and missing links in the CC3M dataset, we train on a CC12M subset of comparable scale, which consists of approximately 3 million valid samples with exact training and validation split sizes on par with the original study.

**ImageNet-100.** The ImageNet-100 dataset is a subset of ImageNet-1K and consists of 100 labels. Training the SAE is performed on 100 randomly selected classes with approximately 1267 images per class (Russakovsky et al., 2015; Tian et al., 2020), accessed through Hugging Face. For experiment analysis[1] and visualization, we use 50 images belonging to the first 100 classes, collected from the ImageNet-100 validation set available on Kaggle.

**CelebA.** The CelebA dataset is a collection of images of celebrities, featuring 20,000 people and 10 images per person. The dataset also includes many attributes, but only use the gender attribute to train a binary classification model (Liu et al., 2015).

**CIFAR10.** The CIFAR10 dataset contains 60,000 color images of 32x32 pixels each. It spans 10 different object categories (Krizhevsky et al., 2009).

**LAION-400m unigrams.** We use the vocabulary from Bhalla et al. (2024), comprising the most frequent unigrams of the LAION-400m captions dataset (Schuhmann et al., 2021)

### 3.3 Hyperparameters

Appendix A presents an overview of the hyperparameters adopted from Zaigrajew et al. (2025), with some modifications applied to our own implementation. The original paper tuned hyperparameters on CLIP RN50

---

[1]The original code repository relies on the Hugging Face implementation of ImageNet-1K, which generates random data splits. This design makes it difficult to recover the corresponding image paths required for our analysis and visualizations. Therefore, we instead use the Kaggle ImageNet-100 dataset, which allows us to manually construct controlled data splits.

before validating on ViT-L/14, while the default `max_nesting` parameter is not explicitly stated. For MSAE, the $k$-list used has powers of 2 from 64 up to the latent dimension.

Our reproduction uses ImageNet-100 instead of ImageNet-1K for validation, the CC12M subset for training and an expansion factor of 8. Due to the default settings in the codebase, the MSAE (RW) model, hereafter referred to as MSAE, was configured with `max_nesting` $= 256$, a $k$-list of $\{64, 128, 256\}$, and $\alpha = \{3, 2, 1\}$. Further exact hyperparameters used in our reproduction for the baseline models are also provided in Appendix A.

### 3.4 Experimental Setup

For our experiments we used the codebase from (Zaigrajew et al., 2025), which is accessible through their GitHub page, and executed the computational tasks on the Snellius HPC. The NVIDIA A100/H100 GPUs were used to run the experiments, with a total computing expense of 4561 SBUs for the reproduction of this study. Resulting in an estimated emission of 6.5 kgCO2eq [2]. We made the following downscaling decisions: (i) we only evaluate the RW weighting approach, as it outperforms UW in the original paper, and (ii) we use only expansion factor 8 (latent dimension of 6144), as higher expansion factors show similar trends.

Reproducing **C1**, we train ReLU SAE ($\lambda \in \{0.001, 0.003\}$), TopK SAE ($k \in \{32, 64\}$) and MSAE to compare the sparsity-fidelity trade-off. For MSAE, we solely use ReLU activation during inference, allowing the model to utilize all neurons it deems necessary for reconstruction. For each model, the progressive recovery is tracked by selecting the top-k activations by magnitude for $k \in \{1, 2, 4, ..., 6144\}$ computing the FVU (reconstruction fidelity) and $L_0$ (sparsity) at each level. Lastly, we evaluate on ImageNet-100 to reproduce the trends from Table 1 as in (Zaigrajew et al., 2025).

For **C2**, we follow the original authors' progressive recovery evaluation, similarly to **C1**. We evaluate the RW version of the MSAE against other baseline models on ImageNet and CC3M. In addition, we test the approach on a simpler dataset, specifically CIFAR10. For each dataset, we encode images into CLIP embeddings using ViT-L/14 and evaluate how well different SAE approaches reconstruct these embeddings as a function of $k$ active latents. We then compare MSAE with TopK and ReLU SAEs. We primarily evaluate based on the following metrics[3]: (i) cosine similarity, (ii) fraction of unexplained variance (FVU), (iii) CKNNA, and (iv) number of dead neurons.

For the experiments used to analyze **C3**, we perform threshold validation, in which 37,445 vocabulary concept neurons are filtered to quantify how many are monosemantic. The original study applies various filter conditions, all of which are outlined in Appendix C. We use the exact same filter conditions, but with a cosine similarity threshold of 0.28. This experiment was performed on the LAION-400m unigrams dataset.

To reproduce **C4**, we repeat two experiments in the original work. These are both examples of how perturbing values of neurons associated with important concepts can lead to meaningful and interpretable changes in CLIP space: First, we train a gender classification model on the CelebA dataset and examine how classification changes as gender-associated neurons are perturbed in SAE-space. Then, we train a nearest-neighbor model on CLIP embeddings to observe how the nearest neighbors change as important concept neurons are perturbed.

---

[2]See Appendix H for carbon emissions calculation.
[3]See Appendix B for metric definitions.

### 3.5 Extension to Original Work

#### 3.5.1 Centroid-Based Cosine Similarity

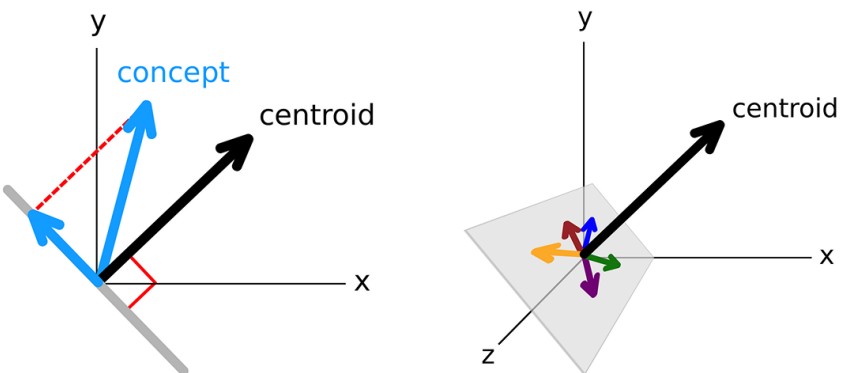

Figure 1: Vocabulary concepts being projected onto the subspace orthogonal to the centroid vector.

We present a more sophisticated approach to validate monosemanticity for each SAE neuron by using a model-agnostic centroid-based method that operates in CLIP space, inspired by Dreyer et al. (2025). The motivation behind this is to provide an alternative to the labor-intensive task of deriving a specific similarity threshold that was used for **C3**.

For a given neuron, we obtain the CLIP image embeddings of the top-$M$ images that most strongly activate for this neuron: $\mathcal{X} = \{x_1, \ldots, x_M\}$ where $x_i \in \mathbb{R}^n$. Next, we compute the centroid by calculating the mean of the image embeddings $\mu = \frac{1}{M} \sum_{i=1}^{M} x_i$ and normalizing it $\hat{\mu} = \frac{\mu}{\|\mu\|}$. Subsequently, we compute the cosine similarities $s \in \mathbb{R}^{|\mathcal{V}|}$ between the centroid $\hat{\mu}$ and each CLIP text embedding $e_c \in \mathbb{R}^n$ for concept $c$ in vocabulary $\mathcal{V}$, where $s_c = \cos(\hat{\mu}, e_c), \ \forall c \in \mathcal{V}$. Finally, we select the top-$K$ concepts with the highest cosine similarity $\mathcal{C} = \text{TopK}_{c \in \mathcal{V}}(s)$. The concepts in $\mathcal{C}$ represent the top concepts most aligned with the centroid. See Appendix I for a pseudocode version.

When the projected concepts collectively describe a single, specific concept, the neuron is classified as monosemantic. In contrast, if a neuron is polysemantic, the average concepts correspond to more general concepts that can be further disentangled. This distinction can be analyzed by observing the angular spread of these concepts. The angular spread is obtained by projecting the average concepts $\mathcal{C}$ onto the subspace orthogonal to the centroid, as illustrated in Figure 1. By examining the angular spread of these projections, we can assess whether a neuron is truly monosemantic, as demonstrated in Section 4.2.1.

#### 3.5.2 Progressive Recovery Capabilities

We largely follow the same methodology as in our reproductions of **C1** and **C2**. To complement the reconstruction curves, we additionally quantify how frequently specific latents are activated across images and perform a qualitative nearest-neighbor analysis in embedding space. For the latter, given an input, we retrieve the dataset image with the highest cosine similarity to its embedding, and then evaluate performance as $k$ increases by checking whether the correct nearest images are identified.

## 4 Results

This section presents the reproducibility results of claims **C1** to **C4**, followed by the results of two smaller extensions. The results generally support all claims, with some exceptions.

### 4.1 Results Reproducing Original Paper

During the validation of **C1**, the objective is to show the same trends as Table 1 of Zaigrajew et al. (2025) could be reproduced. Proving that hierarchical SAE improve the sparsity–fidelity trade-off. To demonstrate this, the emphasis is on comparing the MSAE model to non-hierarchical SAE baselines (e.g., ReLU SAE and TopK SAE) under different configurations. For all models, the relationship between reconstruction fidelity, measured by Fraction of Variance Unexplained (FVU) and sparsity (L0), is illustrated as shown in Figure 2.

Across all of the sparsity–fidelity plots, it can be observed that there is consistent improvement in reconstruction fidelity as sparsity decreases. When comparing the baseline ReLU configurations with $\lambda = 0.001$ and $\lambda = 0.003$, the $\lambda = 0.003$ configuration performs better, achieving lower FVU scores across all sparsity levels compared to the $\lambda = 0.001$ configuration. Another observation emerges when comparing TopK models with $k = 32$ and $k = 64$ activations. The $k = 32$ configuration achieves better reconstruction fidelity at lower $L_0$ values, but plateaus beyond a certain point, whereas the $k = 64$ configuration achieves better reconstruction fidelity at higher $L_0$ values.

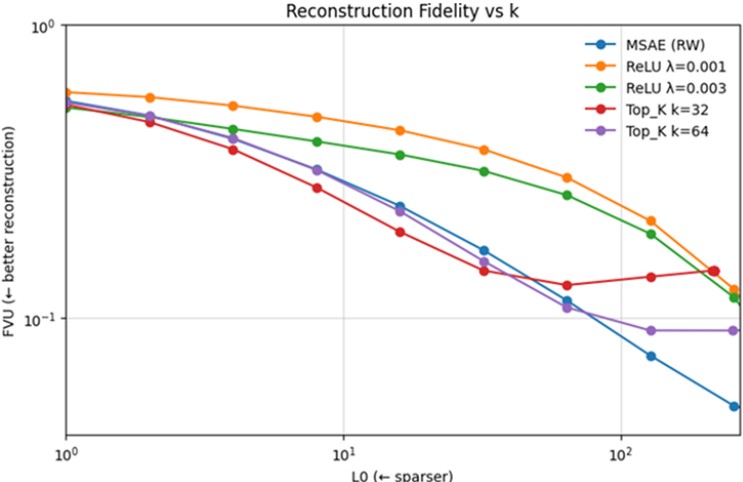

Figure 2: Progressive recovery performance comparing MSAE, ReLU SAE, and TopK SAE on ImageNet-100.

Lastly, it can be validated from Figure 2, as stated by Zaigrajew et al. (2025), that hierarchical sparse autoencoders do improve the sparsity–fidelity trade-off. In the original work the authors demonstrate this using TopK variants with $k = 64$ and $k = 256$ which shows MSAE clearly outperforming the remaining models. In Figure 2 a similar trend can be seen with the exception that the TopK model with $k = 32$ performs better at lower L0 values before plateauing.

For all other configurations, the MSAE model clearly outperforms the non-hierarchical baselines in terms of the sparsity–fidelity trade-off, displaying consistently lower reconstruction error along the progressive recovery curve. Across all settings the same patterns reported in Table 1 of Zaigrajew et al. (2025) are observed: MSAE performs best, followed by the TopK variants, while the ReLU performs worst. In Figure 2 it can also be observed that, once increasing $k$ no longer leads to additional active units, the progressive recovery curves stop.

Despite these differences the relative behavior of the models remains identical; MSAE achieves lower reconstruction error than non-hierarchical baselines at comparable sparsity levels. This indicates that while the values of Table 1 as in Zaigrajew et al. (2025) are not exactly reproduced, the principal sparsity–fidelity relationship described in **C1** is validated.

To validate **C2**, we primarily tried to reproduce Figure 5 from Zaigrajew et al. (2025), as well as Figures 16 and 17 from the original paper's appendix. As mentioned in the analysis of **C1**, the FVU plot in Figure Figure 3 showcases that the reconstruction quality increases with the number of latents. Also, as shown in the original paper, MSAE outperforms the other models. The same applies for the cosine similarity metric.

The plateau for the TopK models (including MSAE) in both metrics is included for completeness and to show the ReLU trends, but it needs to be stated that due to the nature of the TopK procedure, this plateau is expected (since we essentially cut off all latents beyond $k$).

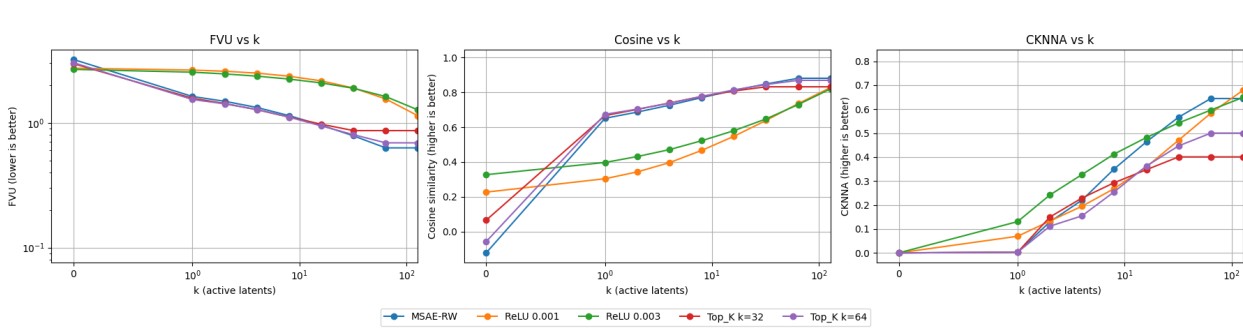

Figure 3: Progressive reconstruction metrics on the ImageNet-100 Dataset.

Interestingly, CKNNA is low for all TopK models (including MSAE) relative to both ReLU models, which is the opposite of what was reported in the original paper. For certain values of $k$, MSAE still outperforms all other model, an effect that becomes even clearer in progressive recovery on other datasets (Figures 12, 13).

To validate **C3**, we aimed to reproduce the original Table 3 from Zaigrajew et al. (2025) for the MSAE model. The original study used 0.42 as the cosine similarity threshold, which was later found to be incorrect[4]. Therefore, we empirically derived a threshold for the distribution in our reproduction study. We first reproduced the maximum similarity distribution for MSAE outlined in Figure 4. We are interested in the sparse concept neurons at the tail that contribute to monosemanticity.

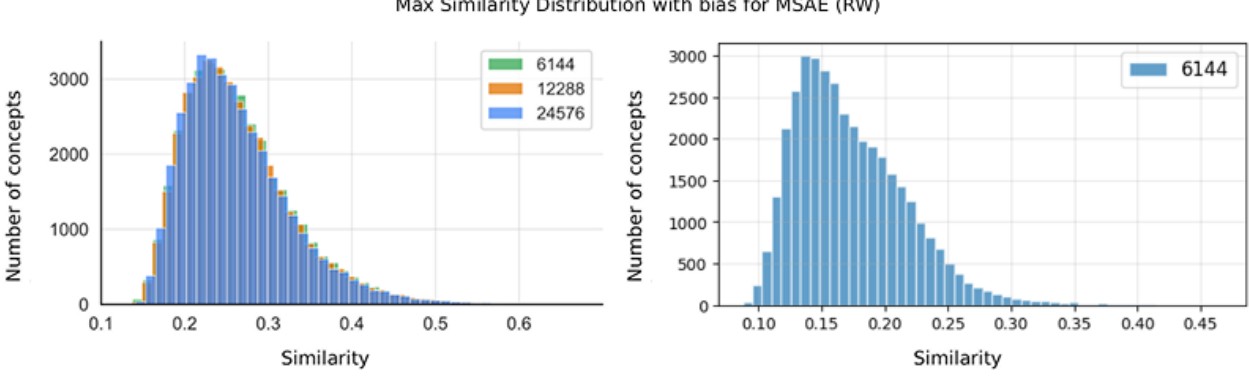

Figure 4: Original (left) vs. reproduced (right) max similarity distribution for the MSAE model.

As outlined in Appendix C, we empirically found that a similarity threshold of 0.28 generalizes across the ReLU, TopK and MSAE models, preserving concept coherence (i.e., all image and text samples share a common concept). The original study demonstrated that sparser architectures (TopK) yield more interpretable concept neurons compared to denser ones (ReLU), with Matryoshka being between the two. When examining the rightmost column in Table 1, we see that the claim also holds for our reproduction study. We manually inspected all 81 concepts of the RW model through their corresponding image and text samples to quantify their monosemanticity. We concluded that only 24 are valid, due to the usage of the ImageNet-100 from Kaggle that only comprises of animals.

---

[4]The authors found a bug in their code related to the preprocessing normalization. After resolving the issue, it resulted in the distribution of our reproduction study. However, the authors did not release a new version of their paper.

|  | Model | Similarity > 0.28 | Best vector | Above and best | Ratio threshold | All conditions |
|---|---|---|---|---|---|---|
| **Reproduced** | ReLU ($\lambda = 0.03$) | 1260 | 1631 | 401 | 194 | 46 |
|  | ReLU ($\lambda = 0.003$) | 308 | 752 | 78 | 133 | 14 |
|  | ReLU ($\lambda = 0.001$) | 51 | 1001 | 14 | 12 | 0 |
|  | TopK ($k = 32$) | 1395 | 1935 | 535 | 663 | 124 |
|  | TopK ($k = 64$) | 1362 | 1891 | 421 | 919 | 144 |
|  | TopK ($k = 128$) | 682 | 1427 | 175 | 832 | 79 |
|  | TopK ($k = 256$) | 370 | 829 | 74 | 381 | 45 |
|  | Matryoshka (RW) | 786 | 1404 | 201 | 774 | 81 |
|  | Matryoshka (UW) | 546 | 1270 | 138 | 590 | 64 |

Table 1: **Comparison of valid concept neurons detected across different SAEs and validation methods.** The validation methods include a cosine similarity threshold above 0.28, selecting the best matching neuron, combining both criteria, applying the concept similarity ratio threshold between the first and second best vocab concept for the neuron, and enforcing all conditions simultaneously. All models used an expansion factor of 8. See Table 5 for a comparison with the original study.

Furthermore, when examining the highest-activating ImageNet-100 images and CC3M texts in Figure 5 for the *trio* concept, we see that the MSAE model has learned monosemantic concepts. See Appendix D for more concepts. However, it is unclear how monosemanticity is defined in the original study. We observe that, despite *trio* being a monosemantic concept, the top-activating images comprise various animals, suggesting that this neuron may be polysemantic and thus can be further disentangled.

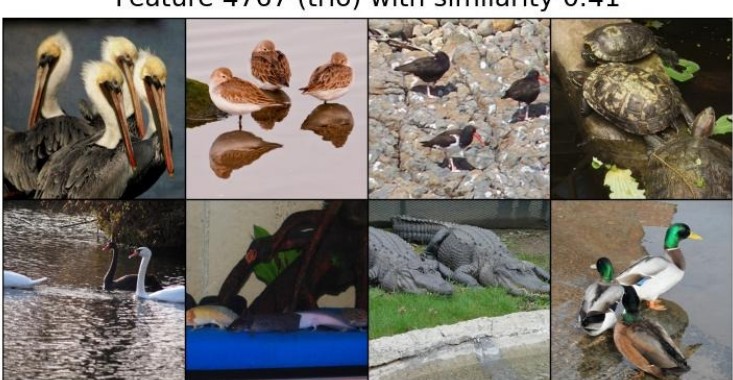

Feature 4767 (trio) with similarity 0.41

1. Three friends toasting and making silly faces at a party
2. Three <PERSON> at the airport
3. Bollywood actor <PERSON> poses with the couple.
4. Three doctors on a pastel background. stock illustration
5. Three of the marker lying next to a notebook for school children, colo...
6. Three students standing on a stage
7. Three empty frames on a wall stock illustration
8. The three kings wise men. Illustration of the three kings wise men roy...

Figure 5: The highest-activating ImageNet-100 images and CC3M texts for the *trio*.

To reproduce **C4**, we repeat two experiments conducted in the original work. Firstly, we train a binary gender classification model on the CelebA dataset. Secondly, we use (k-)nearest neighbor search (NNS) to see how the nearest neighbors in CLIP space change as we amplify neurons corresponding to important concepts. The original work also features a comparison of NNS between SAE and CLIP space, but we found this task of altering CLIP embeddings the most interesting and meaningful in practice, particularly due to the importance of CLIP for inputs to AI systems like OpenAI's Dall-E image generation model (Ramesh et al., 2022).

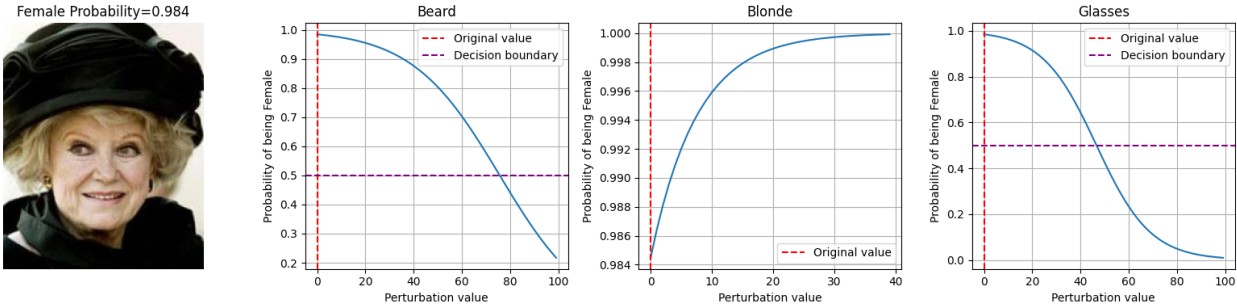

Figure 6: Changes in class probability by increasing MSAE neurons related to certain concepts (*beard, blonde, glasses*). *Beard* and *glasses* biases towards male, while *blonde* biases towards female.

We observe the same biases as those shown in the original work on the gender classification task using the CelebA dataset. Neurons most closely related to the concepts *beard* and *glasses* bias towards a male classification, while the neuron most closely related to *blonde* biases towards a female classification. Additionally, we also find *suit* biases towards male classification and *lipstick* biases towards female classification Appendix F.

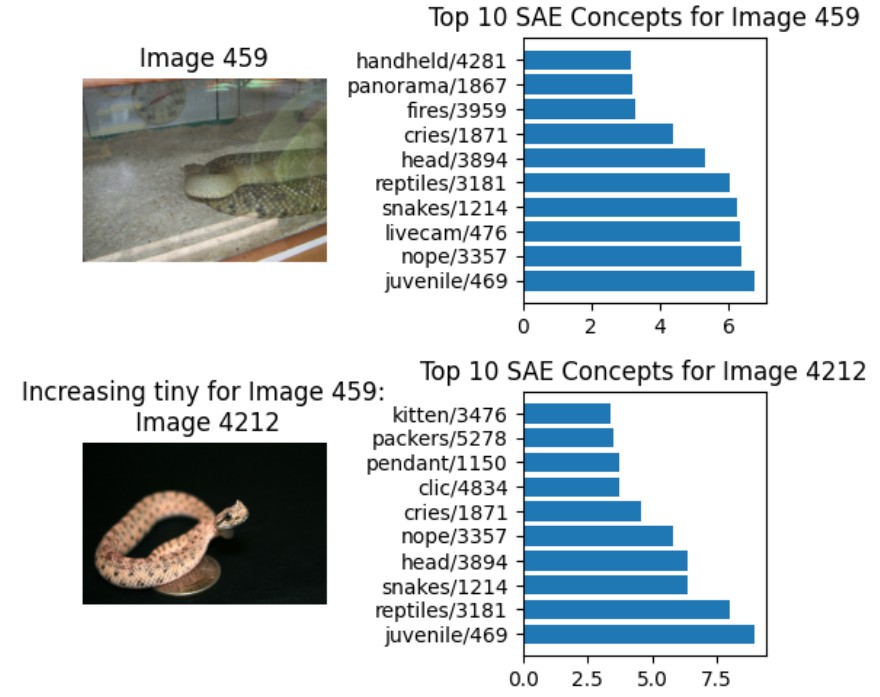

Figure 7: Effect of manipulating MSAE concept on image representations. Transformation from large to tiny.

During the nearest-neighbor experiment, we transform CLIP embeddings to SAE space, change some neurons related to important concepts, and transform back to CLIP space. In the Figure 7, the nearest neighbor of a large snake's CLIP representation becomes a small snake when *tiny* is amplified in SAE space. We refer to Appendix G for more visuals: We observe that a white bird is transformed into a red bird when *red* is increased. One bird transforms into three birds when increasing *trio*, which is similar to what was noted in the original work. To obtain coherent top-$k$ results, the neuron values are set to varying magnitudes.

We consider these experiments to validate **C4**. We found gender biases for the same neurons as in the original work, while also discovering several more. Although our dataset was more limited, the NNS experiment supports that encoding to MSAE latent space allows and for interpretable and monosemantic changes to CLIP vectors.

## 4.2 Results Beyond Original Paper

### 4.2.1 Centroid-Based Cosine Similarity

We observe a single SAE latent neuron for a specific model that is highly associated with the concept *lizards*. When examining Figure 10 for the neuron classified as *lizards*, we find that the top-activating images clearly depict lizards. Consequently, the average concepts most aligned with the centroid are also strongly related to lizards, as shown in Figure 8. We further observe that the projected concept vectors exhibit a dominant angular orientation. From this, we conclude that the *lizards* neuron is monosemantic. In Appendix I we demonstrate a polysemantic case for the concept *trio*.

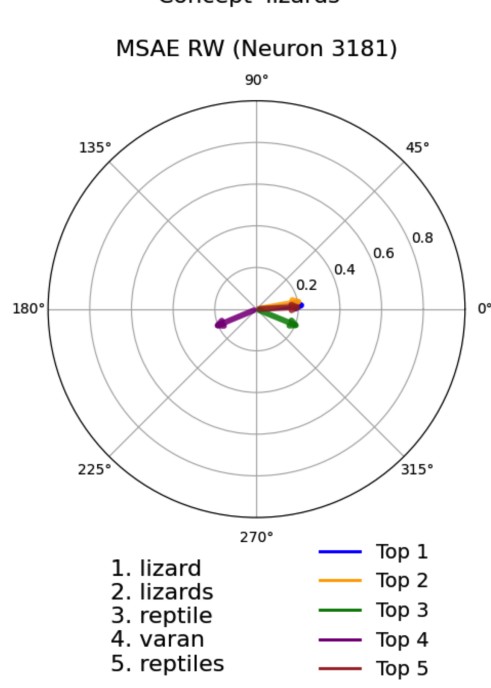

Figure 8: Cosine similarity radial plot for the MSAE model for the neuron highly associated with the *lizards* concept. See Figure 26 for a comparison with multiple models.

### 4.2.2 Progressive Recovery Capabilities

Within the context of progressive recovery, we look at the FVU, cosine similarity, CKNNA and number of dead neurons for progressively higher values of $k$. This is a particularly interesting part of the original paper. For that reason, we qualitatively examined, for different values of $k$, the images closest to the reconstructed embeddings. The goal is to see whether the reconstruction preserves the same semantic content - i.e., whether the nearest neighbors still reflect the intended concept rather than drifting to unrelated images. An example is shown in Figure 9, where we see that for higher $k$ (with the boundary in these two examples being $k = 64$), we consistently obtain the correct image with the highest cosine similarity closest to the input. This also corresponds to Figure 3(b), where the most substantial increase in cosine similarity occurs between the first two values of $k$.

In general, we find that for higher values of $k$, the images were more similar to the inputs, indicating that the reconstructions were visibly better. They also seemed to outperform the other models, especially at lower $k$. In addition to this, in Figure 11, we see that for very large values of $k$, latents are utilized much less than for smaller values of $k$. This quantitatively shows that there can be more separation between the latents, e.g., allowing for specialization and representing specific concepts better, as compared to the overutilized latents for smaller values of $k$.

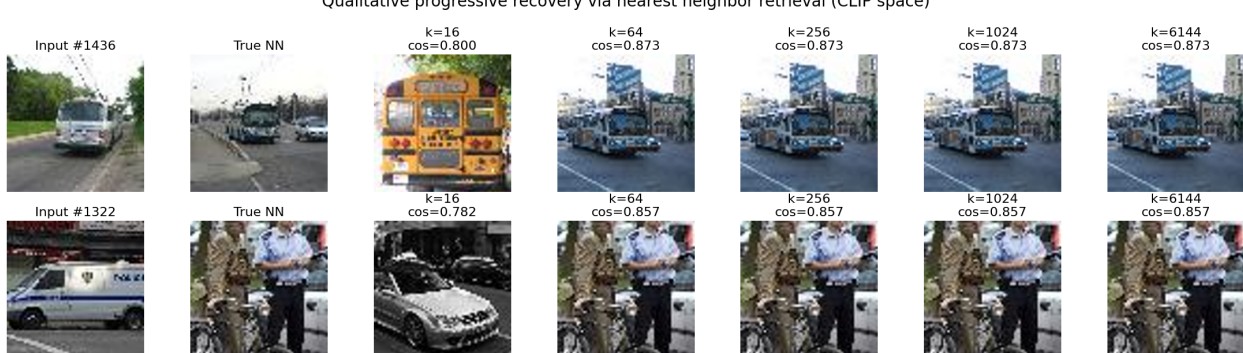

Figure 9: Nearest-neighbor (NN) images to embedding of input image (ImageNet-100)

## 5 Discussion

**Overall reproducibility.** Our experiments aim to reproduce the findings of Zaigrajew et al. (2025) on hierarchical sparse autoencoders for interpreting CLIP embeddings. Overall, we achieve similar reconstruction quality, as well as observe trends comparable to those described in the original paper. However, it is important to understand that there are still some discrepancies - especially in the context of the CKNNA metric and the number of monosemantic concepts.

A key strength of this reproduction is that we fully retrain the models using the original codebase, which is publicly available. Even so, there are several limitations that must also be mentioned. Firstly, we validate on ImageNet-100 rather than ImageNet-1K, which likely affects both neighborhood-based metrics (e.g., CKNNA) and concept discovery rates, and reduces the diversity of visual semantics available for validating discovered concepts. Additionally, we simplified the experimental setup and only focused on the MSAE model, thereby limiting the generalizability.

**C1 - Reproduced:** The progressive recovery curves on ImageNet-100 show that hierarchical training (MSAE) usually achieves lower reconstruction error at comparable sparsity levels than non-hierarchical baselines (e.g., ReLU and TopK SAEs), matching the trend reported in the original work. However, one difference that we observe is that TopK variants can sometimes outperform MSAE, especially at lower $L_0$.

**C2 - Mostly reproduced:** We observe improvements in reconstruction quality as $k$ increases, and qualitative nearest-neighbor analysis shows reconstructions become more faithful with larger $k$. At the same time, we do not fully reproduce all metric-level effects reported by the authors: in our setup, CKNNA is sometimes lower for the MSAE models compared to ReLU baselines, which is the opposite in the original paper. However, we have still managed to find some values where MSAE outperforms the other models. Regardless, this may be due to dataset differences (ImageNet-100) or differences in activation paths, but warrants additional investigation.

**C3 - Mostly reproduced:** After applying the original filtering strategy with an empirically-tuned cosine threshold (0.28 in our setting), we detect a set of concept neurons and recover several monosemantic examples (e.g., *face*, *trio*, *lizards*). However, manual inspection suggests that a large part of the automatically detected concepts is invalid on ImageNet-100, implying that concept discovery is sensitive to the evaluation dataset and to vocabulary coverage. Additionally, a known preprocessing issue in the original code complicates direct numerical comparison with the paper's reported threshold and counts.

**C4 - Reproduced:** Both experiments show that manipulating single concept neurons yields interpretable and systematic changes in downstream behaviour. In the CelebA gender classification setup, perturbing neurons associated with *beard*, *glasses* and *suit* shift predictions toward male, while *blonde* and *lipstick* shift toward female. In nearest-neighbor retrieval, strengthening neurons responsible for abstract properties (size, color and quantity) yields coherent changes in retrieved neighbors (e.g., *large* to *small*, *white* to *red*, *single* to *trio*).

**Future Work.** Our study demonstrated the value in incorporating a centroid-based method to determine monosemanticity for SAE neurons at a more sophisticated level. However, we did not employ this in an automated manner for all latent neurons. Future work can focus on utilizing the angular spread of the concepts, where low spread suggests monosemanticity.

**What was easy:** Running and training the models was straightforward because the authors provided an end-to-end codebase with configs that correspond with paper experiments (albeit there are also some missing links). Additionally, reproducing **C1** and **C2** trends were technically simple, as they rely on standard progressive recovery curves and reconstruction metrics. However, even so, some problems arose due to incomplete methodology explanations in the original paper.

**What was hard:** As mentioned in the previous section, small details about the implementation specifics mattered for accurately reproducing results, e.g., ReLU-only activation paths during inference. Furthermore, the manual nature and sensitivity of the experiments for **C3** proved difficult and time-consuming. In addition, issues such as building a reliable CC3M subset from CC12M and training introduced difficulty, which was time-consuming. We could not find code for the CelebA classification experiment, which is why we used our own.

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

# Appendix

## A   Hyperparameters

| General | | MSAE | |
|---|---|---|---|
| Hyperparameter | Value | Hyperparameter | Value |
| image/text encoder | CLIP ViT-L/14 \| ViT-B/16 | learning rate | $1 \times 10^{-4}$ |
| hyperparameter tuning | CLIP RN50 | $k$ values | $\{64, 128, 256, 512, 1024, 2048, 4096, 6144\}$ |
| training data | CC3M | $\alpha$ (RW) | $\{7, 6, 5, 4, 3, 2, 1\}$ |
| validation data (image) | ImageNet-1K | epochs | 30 |
| validation data (text) | CC3M | batch size | 4096 |
| embedding dim ($n$) | 768 (ViT-L/14), 512(ViT-B/16) | optimizer | Adam |
| expansion factor | 8, 16, 32 | | |
| latent dim (h) | 6144, 12288, 24576 (ViT-L/14), 4096, 8192, 16384 (ViT-B/16) | | |

Table 2: General and MSAE hyperparameters from Zaigrajew et al. (2025).

| TopK SAE | | ReLU SAE | |
|---|---|---|---|
| Hyperparameter | Value | Hyperparameter | Value |
| latent dim ($h$) | 6144 | latent dim ($h$) | 6144 |
| $k$ | $\{32, 64\}$ | learning rate | $5 \times 10^{-5}$ |
| learning rate | $5 \times 10^{-4}$ | epochs | 30 |
| epochs | 30 | batch size | 4096 |
| batch size | 4096 | optimizer | Adam |
| optimizer | Adam | $L_1$ sparsity ($\lambda$) | $\{0.001, 0.003\}$ |

Table 3: TopK and ReLU SAE baseline hyperparameters used in this reproduction study.

# B    Metrics

## B.1    Cosine Similarity (CS)

The cosine similarity metric measures the alignment between two vectors, in this case this would be the original CLIP embedding $x$ and the reconstruction $\hat{x}$:

$$cos(\theta) = \frac{x \cdot \hat{x}}{\|x\|\|\hat{x}\|} \tag{9}$$

with values ranging from 0 (no alignment) to 1 (perfect alignment).

## B.2    Fraction of Variance Unexplained (FVU)

This metric measures the reconstruction fidelity by normalizing the mean squared reconstruction error with the mean squared deviation from the mean (i.e., variance):

$$FVU = \frac{SS_{err}}{SS_{tot}} = \frac{\sum_{i=1}^{N}\|x_i - \hat{x}_i\|^2}{\sum_{i=1}^{N}\|x_i - \bar{x}_i\|^2} \tag{10}$$

where $\bar{x}$ is the mean embedding, and values ranging from 0 (indicating perfect reconstruction) to 1 (predicting the mean).

## B.3    Centered Kernel Nearest Neighbor Alignment (CKNNA)

CKNNA as introduced by Zaigrajew et al. (2025), focuses on the local neighborhood structures after reconstruction. If two samples are close in the original embedding space, they should remain close after reconstruction:

$$\text{CKNNA(K,L)} = \frac{\text{Align(K,L)}}{\sqrt{\text{HSIC(K,K)HSIC(L,L)}}}, \tag{11}$$

$$\text{HSIC(K,L)} = \frac{1}{(n-1)^2}(\sum_i \sum_j (\langle\phi_i, \phi_j\rangle - \mathbb{E}_l[\langle\phi_i, \phi_l\rangle])(\langle\psi_i, \psi_j\rangle - \mathbb{E}_l[\langle\psi_i, \psi_l\rangle])), \tag{12}$$

$$\text{Align(K,L)} = \frac{1}{(n-1)^2}(\sum_i \sum_j \alpha(i,j)(\langle\phi_i, \phi_j\rangle - \mathbb{E}_l[\langle\phi_i, \phi_l\rangle])(\langle\psi_i, \psi_j\rangle - \mathbb{E}_l[\langle\psi_i, \psi_l\rangle])),$$
$$\alpha(i,j;k) = 1[i \neq j \text{ and } \phi_j \in \text{knn}(\phi_i;k) \text{ and } \psi_j \in \text{knn}(\psi_i;k)] \tag{13}$$

where $\phi$ is the original CLIP embedding and $\psi$ being the SAE activations. The indicator function $\alpha(i,j;k)$ returns 1 when: (i) the two samples are not the same ($i \neq j$) (ii) sample $j$ is part of the $k$-nearest neighbors of sample $i$, in original embedding space ($\phi_j \in \text{knn}(\phi_i;k)$) (iii) sample $j$ is part of the $k$-nearest neighbors of sample $i$, in SAE activation space ($\psi_j \in \text{knn}(\psi_i;k)$).

The Align function calculates the similarity across the neighbor pairs, while HSIC calculates the overall similarity between the two spaces. By dividing Align with HSIC the score becomes normalized, where higher values indicate that local neighborhoods are better preserved.

## B.4    Number of Dead Neurons (NDN)

NDN measures how many neurons remain inactive (zero) in the SAE activation layer, counted across all inputs during training/evaluation. A high value indicating poor utilization of neurons, while a lower value indicates better utilization.

## C    Concept Discovery and Validation

To address the challenges for **C3** outlined in Section 3.4, we applied the three validations outlined in the original study, but used a cosine similarity threshold of 0.28 instead of the original threshold (0.42):

1. Cosine similarity $> 0.28$, which ensures that the neurons exhibit strong alignment with their assigned concepts, preventing weak or ambiguous concept mappings.

2. Concept similarity ratio $\frac{\text{Top similarity}}{\text{Second-highest similarity}} > 2.0$, confirms concept uniqueness by requiring the best match to be at least twice as strong as the second-best concept, avoiding distributed representations.

3. One concept per neuron (with the highest similarity) enforces monosemanticity by assigning only the most strongly aligned concept to each neuron, which is needed due to the vocabulary structure containing multiple variations of the same concept (e.g., 'bird' and 'birdie').

The cosine similarity threshold makes this experiment rather brittle, since this number depends on the training dataset, the vocabulary used to compute the cosine similarity matrix, and the choice of model. Deriving this threshold is a labor-intensive task of manually examining individual concepts for multiple models by adjusting the threshold and see where coherence among the image and text samples is lost.

The similarity threshold of 0.28 used in our reproduction study was determined by manually examining the minimum value required to preserve concept coherence across both text and image samples. To identify a threshold that generalizes across models, we first collected the top-2 highest concept thresholds for each concept (shown in bold in Table 4) and then selected the minimum value among them. From this analysis, we derived a threshold of 0.28 that maintains concept coherence for most models.

| Concept | ReLU | TopK | MSAE |
|---------|------|------|------|
| face    | 0.31 | 0.15 | **0.28** |
| trio    | **0.34** | 0.42 | 0.42 |
| smiles  | 0.24 | **0.32** | 0.33 |
| birds   | 0.13 | 0.32 | **0.31** |
| lizards | 0.11 | 0.30 | **0.29** |
| duo     | **0.26** | 0.28 | **0.26** |

Table 4: Concept similarity scores across SAE models. Bold values indicates the top-2 distinct similarity threshold among the models for a given concept. Any lower value of at least 0.1 exhibits a significant loss in concept coherence.

By applying the previously mentioned filter conditions on the vocabulary concept neurons, we produced Table 5. Below is the full version that also compares our reproduced version to the original study's result.

| | Model | Similarity > 0.28 | Best vector | Above and best | Ratio threshold | All conditions |
|---|---|---|---|---|---|---|
| **Reproduced** | ReLU ($\lambda = 0.03$) | 1260 | 1631 | 401 | 194 | 46 |
| | ReLU ($\lambda = 0.003$) | 308 | 752 | 78 | 133 | 14 |
| | ReLU ($\lambda = 0.001$) | 51 | 1001 | 14 | 12 | 0 |
| | TopK ($k = 32$) | 1395 | 1935 | 535 | 663 | 124 |
| | TopK ($k = 64$) | 1362 | 1891 | 421 | 919 | 144 |
| | TopK ($k = 128$) | 682 | 1427 | 175 | 832 | 79 |
| | TopK ($k = 256$) | 370 | 829 | 74 | 381 | 45 |
| | Matryoshka (RW) | 786 | 1404 | 201 | 774 | 81 |
| | Matryoshka (UW) | 546 | 1270 | 138 | 590 | 64 |
| | **Model** | **Similarity > 0.42** | **Best vector** | **Above and best** | **Ratio threshold** | **All conditions** |
| **Original** | ReLU ($\lambda = 0.03$) | 3308 | 2740 | 874 | 380 | 97 |
| | ReLU ($\lambda = 0.003$) | 896 | 2372 | 217 | 395 | 29 |
| | ReLU ($\lambda = 0.001$) | 351 | 4116 | 77 | 169 | 8 |
| | TopK ($k = 32$) | 4081 | 2755 | 1021 | 999 | 216 |
| | TopK ($k = 64$) | 3797 | 2557 | 873 | 1322 | 238 |
| | TopK ($k = 128$) | 2141 | 2167 | 455 | 1508 | 211 |
| | TopK ($k = 256$) | 943 | 1883 | 168 | 1579 | 134 |
| | Matryoshka (RW) | 1136 | 1628 | 237 | 1429 | 140 |
| | Matryoshka (UW) | 907 | 1517 | 195 | 1254 | 125 |

Table 5: **Full comparison of valid concept neurons detected across different SAEs and validation methods.** The validation methods include a cosine similarity threshold above 0.28 for our reproduction study and 0.42 for the original study, selecting the best matching neuron, combining both criteria, applying the concept similarity ratio threshold between the first and second best vocab concept for the neuron, and enforcing all conditions simultaneously. All models used an expansion factor of 8.

# D Interpreting CLIP with MSAE: Additional Results

### Feature 3494 (face) with similarity 0.28

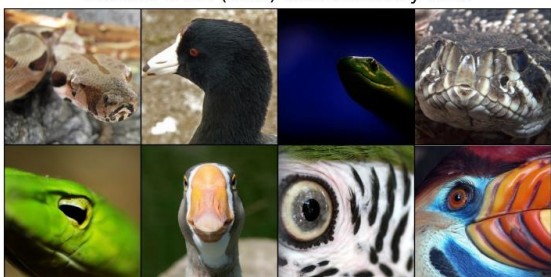

1. Even These Lips Have Brows is listed (or ranked) 8 on the list The 24 ...
2. Before And After Pictures Show How A Nose Job Can Change Your Face
3. Head of the <PERSON> from a 3d Grid. <PERSON> Model. Face Scanning. Vi...
4. Sketch of the girl's face royalty free illustration
5. Head of a Woman in a Night Cap by <PERSON> - Portrait Drawings from He...
6. Vector image of an cheetah face. On black background royalty free illu...
7. Delivery Human Head Logo Icon Design. This design can be used as a log...
8. Head of an old asian men, cartoon isolated vector illustration. Set of...

### Feature 2230 (birds) with similarity 0.33

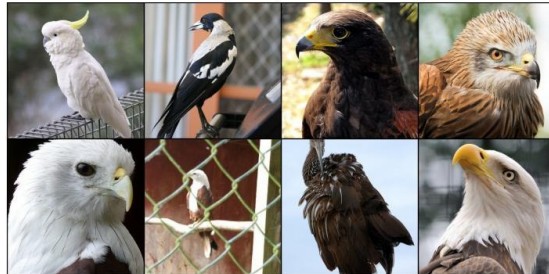

1. Help Build a Free Flight Aviary for Rescued Birds
2. Crow bird, poultry animal silhouette. Good use for symbol, logo, web i...
3. A photo of <PERSON> holding a fake bird.
4. Bird - Both seats are occupied
5. The Bird Alphabet Book | Children's Book | Learn the Alphabet with Bir...
6. The virtual birds stock illustration
7. Negative space bird of the abstract circle
8. <PERSON>'s Nest in a Tree Poster - Birds

### Feature 3181 (lizards) with similarity 0.32

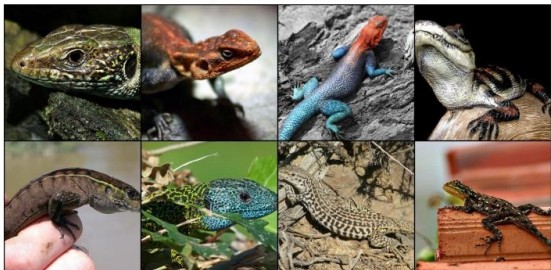

1. Water Monitor big lizard on a coconut tree near the water in Thailand.
2. Crocodilians The largest living reptiles.
3. Lizards generally live on a diet of insects, including plant-eaters, l...
4. Crocodile: portrait on a black
5. Kraków, Wawel Dragon, Poland, the statue, tourism, symbol, HD wallpape...
6. Roaming Reptiles, Parties with a difference!
7. Bearded Dragon smile is the most adorable thing. Reptiles And Amphibia...
8. The Lizard, <PERSON>, <PERSON>, Amphibians, Nature, Animals

### Feature 3938 (duo) with similarity 0.26

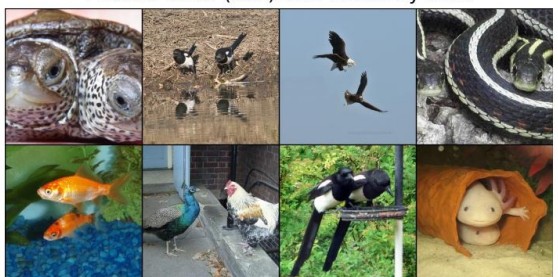

1. A pair of Developers working at a computer
2. Photo of two men on a railway line
3. Two men smiling after their celebration at The Gables in Malvern
4. Two men onstage presenting an award.
5. An old man and a boy
6. Two mountain bikers looking out over a mountain-top view
7. 2 brothers stealing sweets in the kitchen - Stock Image
8. Two men talking at the Jazz Reception

Figure 10: The highest-activating ImageNet-100 images and CC3M texts for more MSAE concepts

# E   Progressive Recovery: Additional Results

This appendix introduces more results for progressive recovery on different datasets: CIFAR10 and CC3M. It also contains a plot for latent utilization.

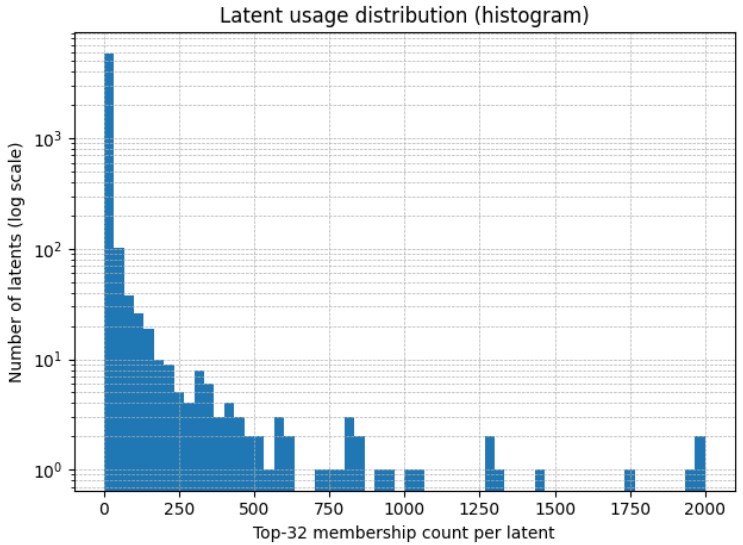

Figure 11: Distribution of latent usage with different values of $k$

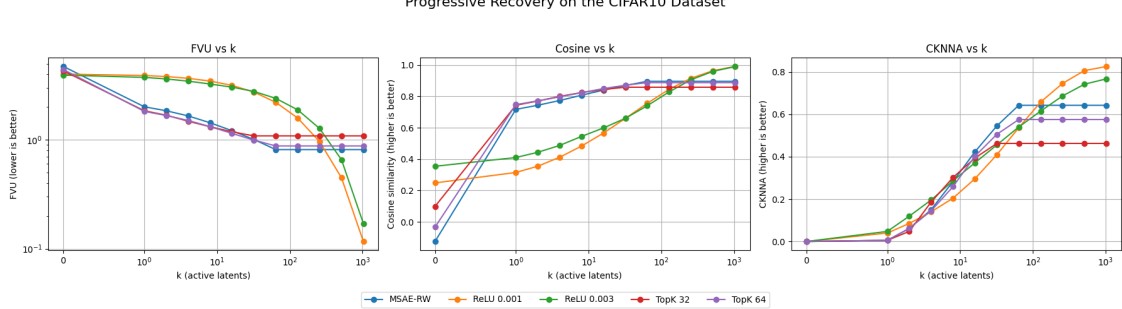

Figure 12: Progressive recovery metrics on CIFAR10 dataset

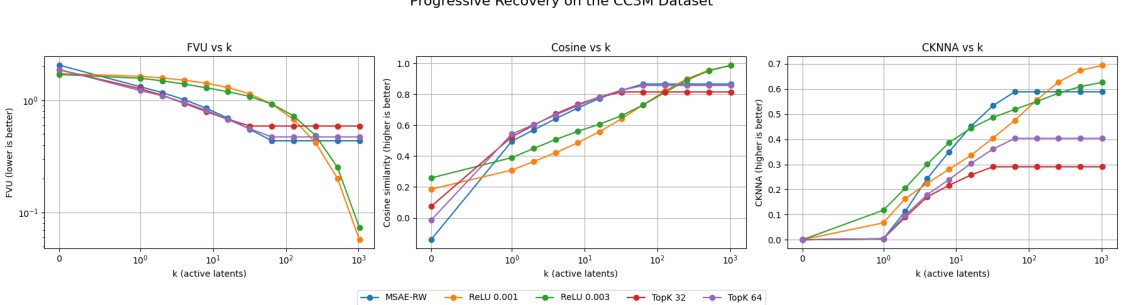

Figure 13: Progressive recovery metrics on CC3M dataset

# F    CelebA Gender Classification Experiment

This Appendix section concerns the CelebA gender classification experiment. We include additional findings: suit biases towards male classification and lipstick biases towards female classification. The results are also validated by experimenting on a different (male class) image, which shows the same biases. We also display the images with the highest activations for each of the investigated neurons.

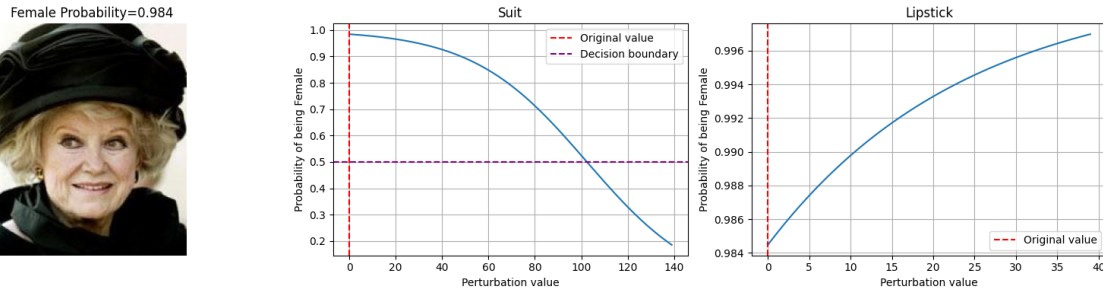

Figure 14: Neurons that are most closely associated with the suit and lipstick concepts also display gender bias: suit biases towards male classification and lipstick biases towards female classification

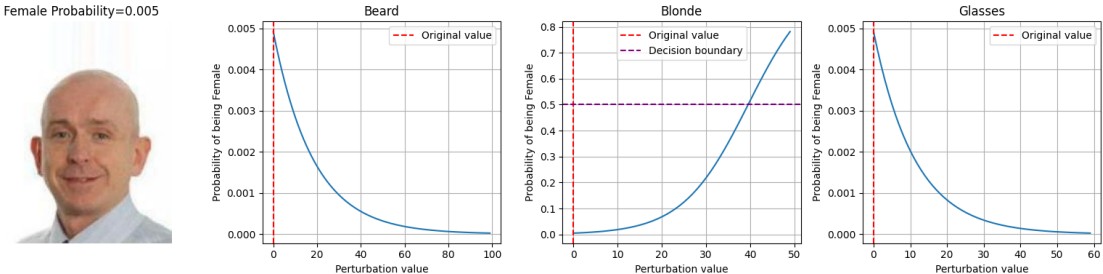

Figure 15: Additional experiment that validates the findings of biases in the neurons most closely associated with *beard* (male bias), *blonde* (female bias) and *glasses* (male bias).

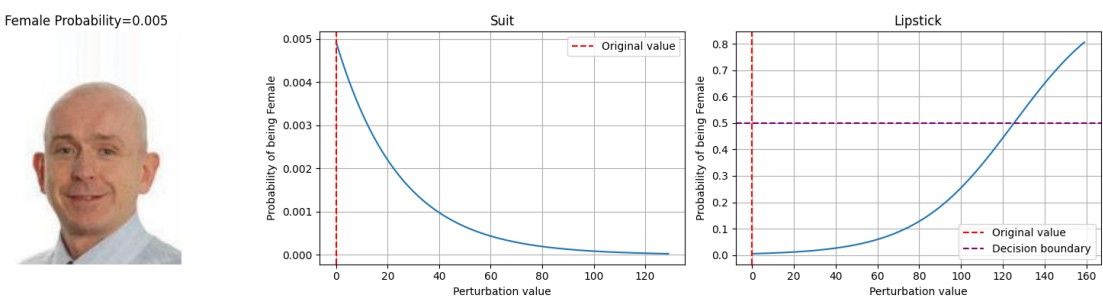

Figure 16: Additional experiment that validates the findings of biases in the neurons most closely associated with *suit* (male bias) and *lipstick* (female bias).

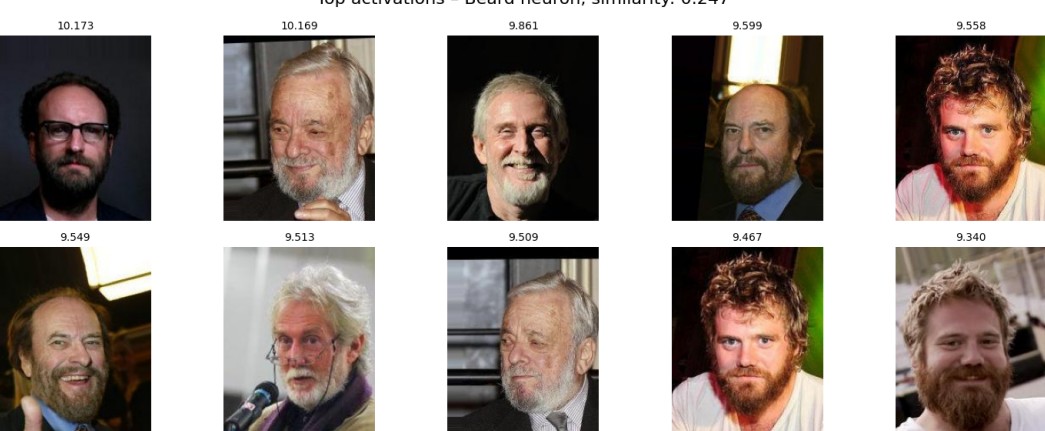

Figure 17: Images with the highest activation for the neuron most closely corresponding to *beard*

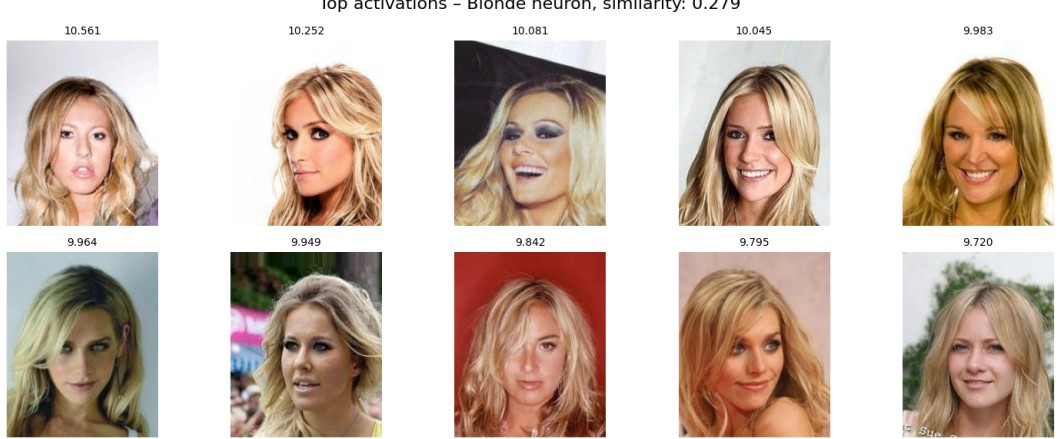

Figure 18: Images with the highest activation for the neuron most closely corresponding to *blonde*

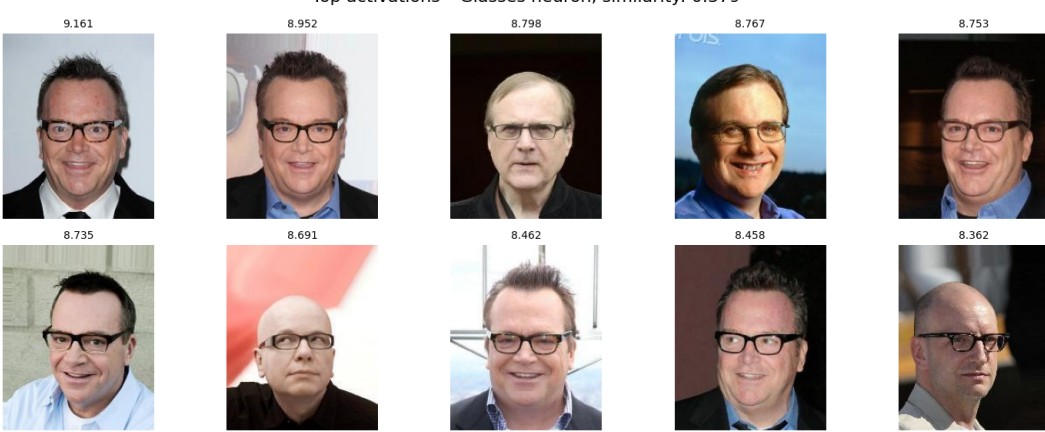

Figure 19: Images with the highest activation for the neuron most closely corresponding to *glasses*

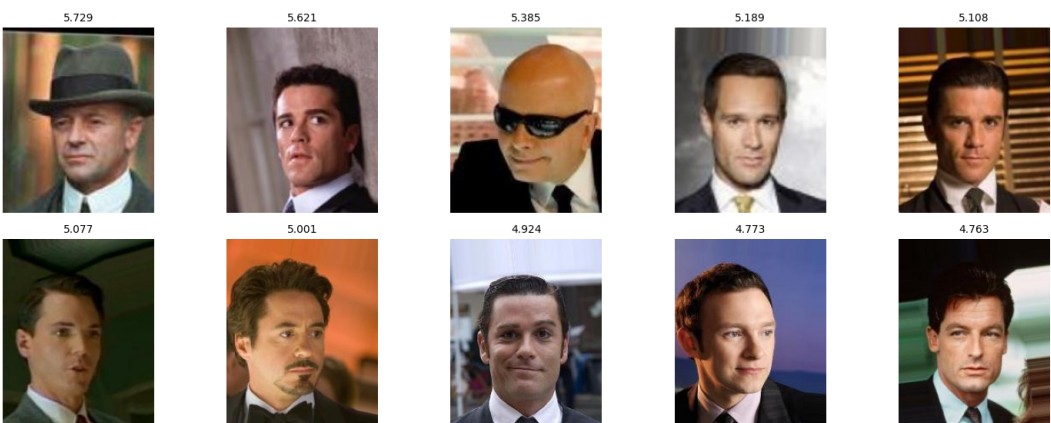

Figure 20: Images with the highest activation for the neuron most closely corresponding to *suit*

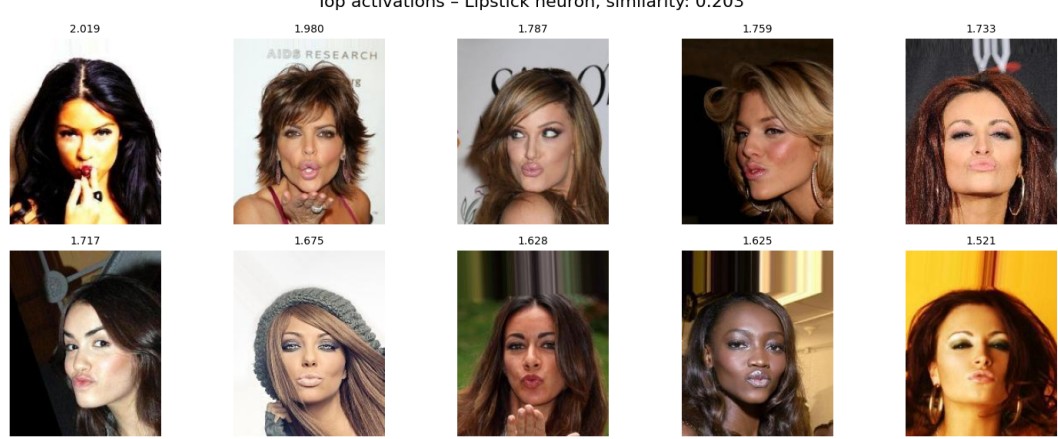

Figure 21: Images with the highest activation for the neuron most closely corresponding to *lipstick*

## G   Imagenet Nearest Neighbors Experiment

We show top-3 results below. Our findings suggest that in order to show the most consistent top-3 results the perturbations have to be of varying magnitudes, however, 100 appears to be a good baseline.

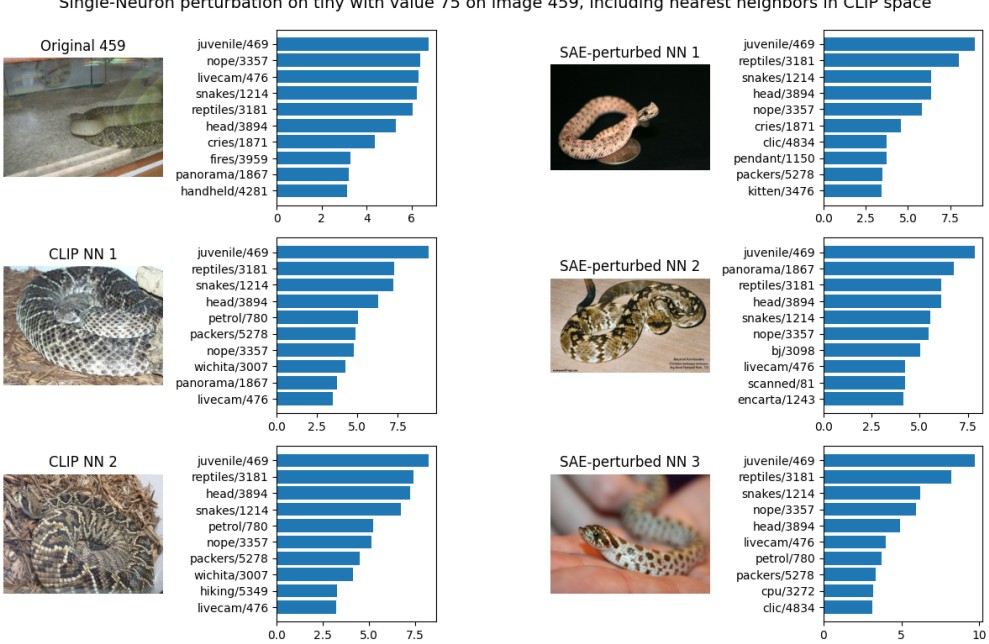

Figure 22: Large snakes transform into primarily small snakes when the neuron most closely related to the concept *tiny* is increased.

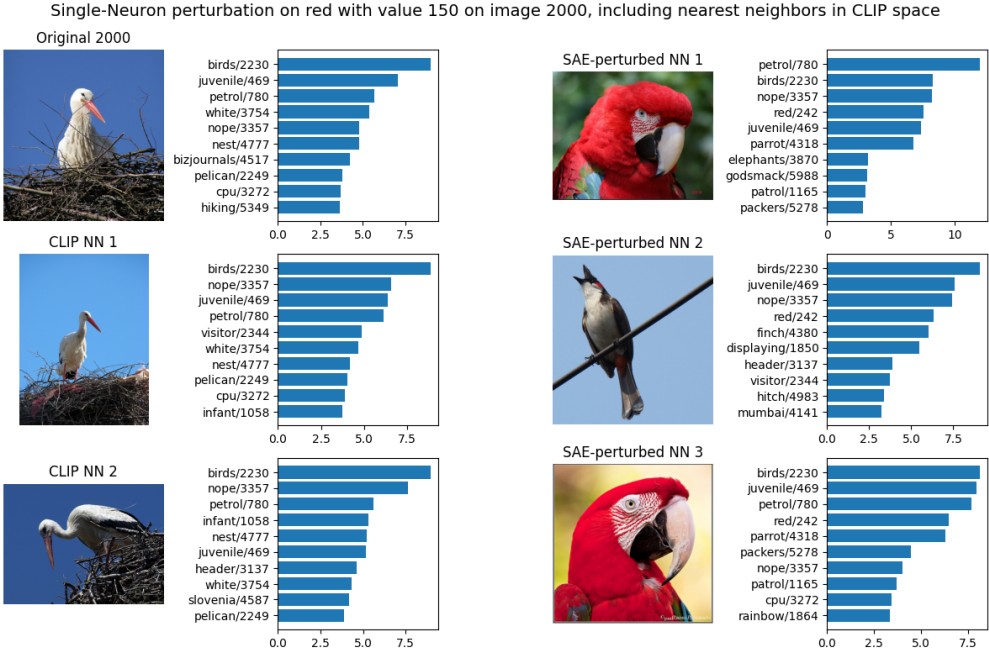

Figure 23: When increasing the neuron associated with *red*, the original image and its nearest neighbors transition away from (white) pelicans and become red birds (primarily parrots.)

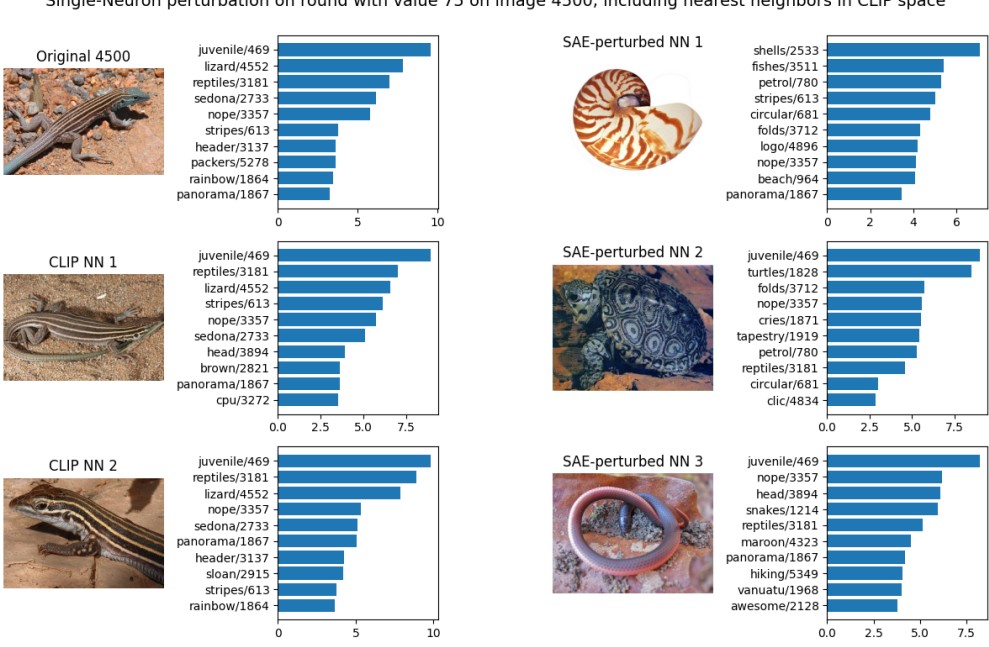

Figure 24: The neuron most closely corresponding to *round* is increased by 75. The original image and its nearest neighbors are no longer lizards, but instead animals with the shared characteristic *round* are shown.

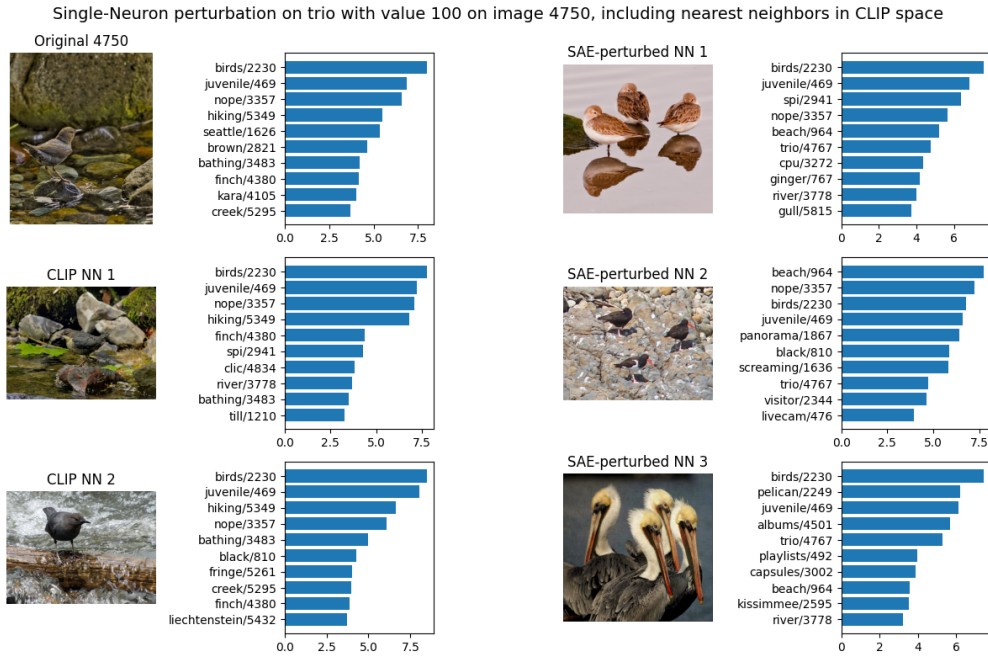

Figure 25: When, the neuron most closely corresponding to *trio* is increased by 100, we observe that that the nearest neighbors which were originally single birds turn into 3 birds, or 4 in the last neighbor.

## H    Carbon Emissions Calculation

Carbon emissions were calculated as follows:

$$\text{kgCO}_2\text{eq} = \text{Activity Data} \times \text{Emission Factor} \tag{14}$$

where the activity data represents the total energy consumption in kWh calculated by converting Service Billing Units (SBUs) to GPU hours and multiplying it by the average power consumption:

$$\text{Activity Data} = \frac{\text{SBU}_{total}}{\text{SBU}_{\text{avg}}} \times \text{Power}_{\text{avg}} \tag{15}$$

The average SBU rate across A100 and H100 partitions is $\text{SBU}_{\text{avg}} = (128 + 192)/2 = 160$ SBU/hour (SURF, 2025), and the average power consumption is $\text{Power}_{\text{avg}} = (0.4 + 0.7)/2 = 0.55$ kW (NVIDIA, 2024a;b). The emission factor is the Dutch carbon intensity for January 2026: Emission Factor $= 0.415$ kgCO2eq/kWh (Nowtricity, 2025).

# I Extension 1: Additional Results

---

**Algorithm 1** Pseudocode for the centroid-based cosine similarity method for a neuron

---

**Require:** Neuron index $n \in \{1, \dots, D\}$, top-$M$ activating images, vocabulary $\mathcal{V}$
**Ensure:** Top-$K$ average concepts $\mathcal{C}$ associated with neuron $n$
 1: Obtain CLIP image embeddings $\mathcal{X} = \{x_1, \dots, x_M\}, \quad x_i \in \mathbb{R}^d$ from the top-$M$ activating images
 2: Compute the centroid of image embeddings: $\mu \leftarrow \frac{1}{M} \sum_{i=1}^{M} x_i$
 3: Normalize the centroid: $\hat{\mu} \leftarrow \frac{\mu}{\|\mu\|}$
 4: Initialize similarity vector $s \in \mathbb{R}^{|\mathcal{V}|}$
 5: **for** each concept $c \in \mathcal{V}$ **do**
 6:     Retrieve CLIP text embedding $e_c \in \mathbb{R}^d$
 7:     Compute cosine similarity: $s_c \leftarrow \cos(\hat{\mu}, e_c)$
 8: **end for**
 9: Select top-$K$ average concepts: $\mathcal{C} \leftarrow \text{TopK}_{c \in \mathcal{V}}(s)$

---

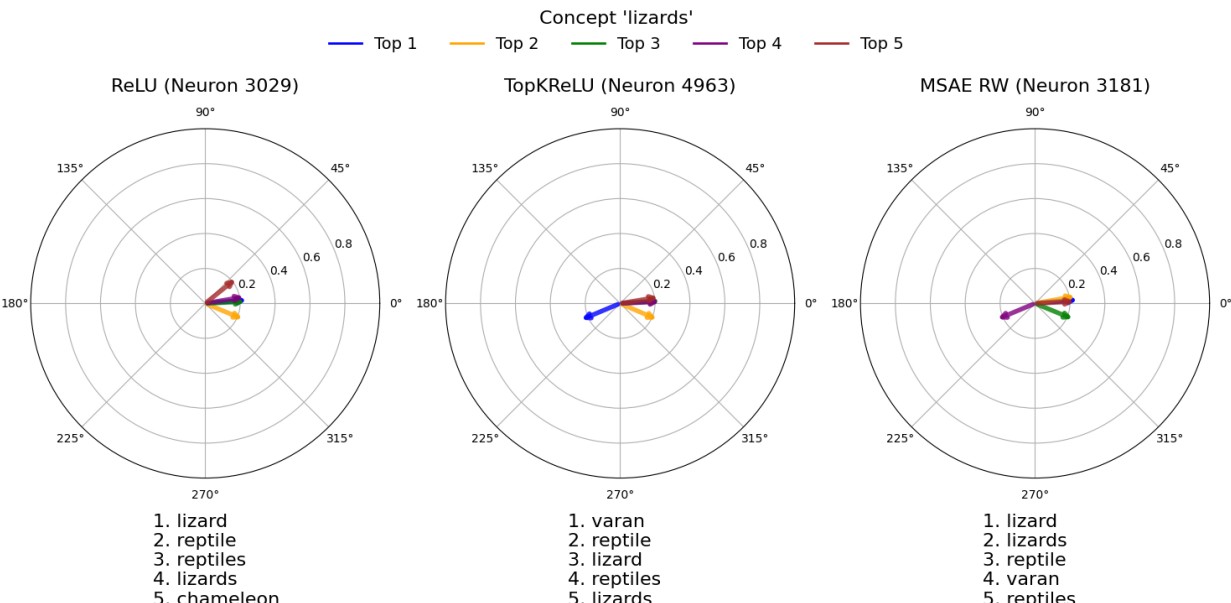

Figure 26: Cosine similarity radial plot for the ReLU ($\lambda = 0.003$), TopKReLU ($k = 64$), and MSAE models. The radius corresponds to the cosine similarity with the centroid, while the angle represents the direction of the projection within the 2D subspace orthogonal to the centroid vector. The top-5 average concepts most aligned with the centroid represent the *lizards* concept. Together with the observation that most projections exhibit a similar angular orientation, we conclude that the neuron classified as *lizards* in each model is monosemantic.

We examine a neuron that is highly associated with the concept *trio* in a given model. When examining the top-activating images for this neuron, shown in Figure 5, we observe images containing various animals rather than a single, consistent concept. Correspondingly, the top-5 concepts most aligned with the centroid (Figure 27) do not reflect a specific animal category but instead comprise more general concepts (e.g., trio, triplets, breeding). This indicates that the *trio* neuron is polysemantic. This polysemanticity is further supported by the angular spread of the projected concepts, which does not concentrate in a single direction.

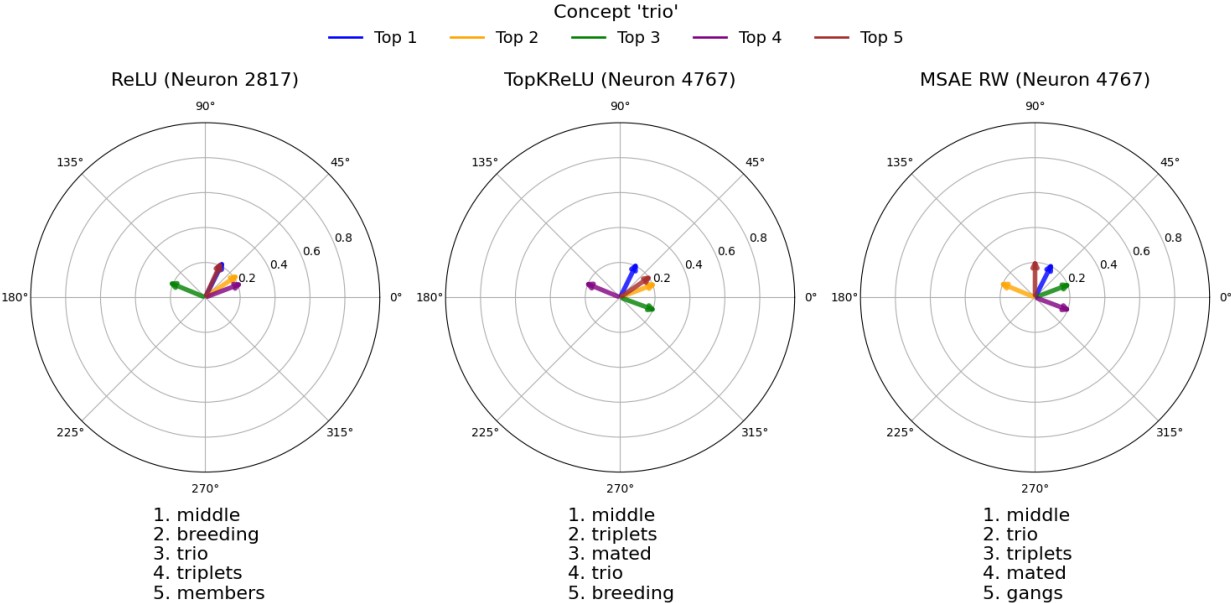

Figure 27: Cosine similarity radial plot for the ReLU ($\lambda = 0.003$), TopKReLU ($k = 64$), and MSAE models. The radius corresponds to the cosine similarity with the centroid, while the angle represents the direction of the projection within the 2D subspace orthogonal to the centroid vector. The top-5 average concepts most aligned with the centroid comprise more general concepts. Together with the observation that the projections do not concentrate in a single direction, we conclude that the neuron classified as *trio* in each model is polysemantic.

