# OpenReview forum: "[Re] Interpreting CLIP with Hierarchical Sparse Autoencoders"
_TMLR — Rejected by TMLR_

### Review · Reviewer_fcug · 2026-03-20

**Summary Of Contributions:**

This paper aims to reproduce and extend the experiments of Matryoshka Sparse Autoencoders (MSAE). Specifically, the authors focus on the reproducibility of the following claims: MSAE improves the sparsity-fidelity tradeoff (C1), MSAE enables hierarchical and progressive representations (C2), MSAE discovers monosemantic and interpretable concepts (C3), and MSAE can be used for concept intervention (C4). The authors report that they were largely successful in reproducing the original results.
In addition, they introduce a tool based on centroid-based cosine similarity designed to assess the monosemanticity of neurons in an automated fashion.

Strengths:
- The writing is of high quality. The experiments are extensive and the experimental setup is clearly described.
- The motivation is highly welcomed, particularly in a context where incentives strongly favor the creation of new methods over their reproduction. There is a genuine issue with the robustness of the foundations of deep learning research, and this work contributes meaningfully to addressing it. I also want to highlight that this type of article is especially relevant for TMLR, given the journal's emphasis on scientific rigor over novelty.
- The choice of paper to reproduce is well-justified. Sparse autoencoders have attracted considerable hype recently, despite facing criticism, and this work helps to ground the discussion about their relevance.
- The authors succeed in identifying the limitations of the original paper.
- The inclusion of a carbon footprint estimate is a commendable effort.
- The Centroid-Based Cosine Similarity tool is a creative and visually interpretable approach to tackling polysemanticity. If proven viable, it has the potential to become a widely adopted tool in mechanistic interpretability. The application presented in Section I is particularly interesting.
- The analyses provided for C3 and C4 are thorough and balanced.

Weaknesses:
- There is, overall, a noticeable gap between the results presented and their interpretation. Looking at the results — particularly for C1 and C2 — I would be less inclined to conclude that reproducibility was achieved. While the discussion section partially addresses this, I think the authors could lean more explicitly toward characterizing the reproducibility as mixed, especially in the abstract.
- The codebase used is the same as in the original paper. As a consequence, this study does not cover the possibility of coding mistakes in the original implementation.
- The interpretation of Figure 1 is somewhat unclear: MSAE appears to be outperformed by Top-K in most regimes, which seems to contradict the original experiments — yet the authors still conclude that C1 is valid.
- Regarding C2, my reading of Figure 3 does not support the conclusion that MSAE outperforms other methods; at best, it suggests competitive performance. More critically, the CKNNA results are opposite to those reported in the original paper, yet no further comment is provided and the authors continue to lean toward validating C2 despite this discrepancy — why is this labeled as "Mostly reproduced"?
- I also have concerns about the decision to not explore certain experimental setups (see requested changes for details).
- Some plots are difficult to read (see requested changes for concrete examples).
- While the experiment presented in Section I is interesting, some results are puzzling. There is also some ambiguity around the magnitude of the vectors, which appear very close to one another.

**Audience:**

Yes

**Audience Explanation:**

The paper targets the reproduction of a work that sits at the center of significant interest and scrutiny. Given the broad audience that the original paper attracts, this reproduction constitutes a valuable read for the community.

**Claims And Evidence:**

Yes

**Claims Explanation:**

While I have some concerns regarding the interpretation of the results and the lack of argumentation in cases where they diverge from the original paper, the experiments are overall extensive, pertinent, and appear to be accurate.

**Requested Changes:**

Major:
- Lower the reproducibility claims, particularly in the abstract and throughout the analysis of results.
- The statement "We only evaluate the RW weighting approach, as it outperforms UW in the original paper" appears only partially accurate — for instance, L0 is generally higher under RW. This choice depends on how one weighs sparsity against reconstruction in the tradeoff. Consider including uniform weighting, at least for C1.
- Still regarding C1, why did the authors not investigate lower values of k, which appear to correspond to a more competitive regime?
- Regarding Section I: can the authors explain why some words have different angles yet similar meanings (e.g., "varan" vs. other concepts in the middle of Figure 26)? Can the authors also confirm that there is a statistically significant decrease between the top concepts and the others?
- Figure 3 is unclear. The objective is to reproduce Figure 5 of the original study, but the x-axis appears to represent k, whereas the original article uses L0. Can the authors clarify this discrepancy?

Minor:
- The graduations in Figure 1 are difficult to read. Please add clearer graduations, and reconsider the logarithmic y-axis. Similar issues apply to Figures 26 and 27, where the magnitude is hard to interpret.

Very Minor (concerns noted, but not blocking):
- Only an expansion factor of 8 is evaluated. While the authors argue that higher expansion factors show similar trends, this remains a mild disappointment given that it is a pivotal hyperparameter in SAEs.

---

### Review · Reviewer_Smky · 2026-03-29

**Summary Of Contributions:**

## Summary

This paper presents a reproduction study of *Interpreting CLIP with Hierarchical Sparse Autoencoders* (Zaigrajew et al., 2025), focusing on four central claims: (C1) improved sparsity–fidelity trade-off, (C2) hierarchical/progressive representations, (C3) discovery of monosemantic concepts, and (C4) concept-based interventions.

Overall, I think the study successfully reproduces the main qualitative trends reported in the original work. In particular, the authors provide convincing evidence that hierarchical SAEs (MSAE) generally achieve better sparsity–fidelity trade-offs and enable meaningful concept-based manipulations. At the same time, the paper identifies several discrepancies, including differences in neighborhood-based metrics (CKNNA), sensitivity of monosemanticity to thresholding, and preprocessing inconsistencies in the original implementation.

The inclusion of extensions, such as the centroid-based method for assessing monosemanticity, is well motivated and aligned with the goals of the original work.

---

## Strengths

- **Comprehensive coverage of core claims.**
  The paper evaluates all major claims of the original work (C1–C4), spanning reconstruction, hierarchy, interpretability, and interventions. This makes the reproduction feel complete rather than selective.

- **Careful identification of inconsistencies.**
  One of the strongest aspects of this work is that it does not simply confirm results, but also highlights important issues such as preprocessing bugs, threshold mismatches, and deviations in CKNNA behaviour. I think this adds real value beyond replication.

- **Transparent reporting of experimental deviations.**
  Differences in datasets (e.g., ImageNet-100 vs ImageNet-1K), scaling, and hyperparameters are clearly documented, which makes it easier to interpret the results.

- **Combination of quantitative and qualitative evaluation.**
  The inclusion of nearest-neighbour analysis, manual inspection of concepts, and intervention experiments provides useful insight beyond standard reconstruction metrics.

- **Well-motivated and relevant extensions.**
  The centroid-based monosemanticity method directly addresses a weakness in the original evaluation and is conceptually well aligned.

- **Effectively leverages structure in pretrained representations.**
  An important strength of the method, which also comes through in this reproduction, is its ability to extract and reorganize semantic structure already present in CLIP embeddings into sparse and interpretable components. This suggests that the approach is particularly effective when the underlying representation space exhibits strong semantic separability.

- **Useful discussion of practical reproducibility challenges.**
  The discussion of what was easy versus difficult to reproduce (e.g., missing implementation details, dataset construction issues) is informative and helpful for the community.

---

## Weaknesses

- **No independent reimplementation.**
  A key limitation is that the study relies on retraining the original codebase rather than reimplementing the method. As a result, the paper demonstrates robustness of the implementation, but provides more limited evidence for full methodological reproducibility.

- **Limited comparability due to dataset substitutions.**
  The use of ImageNet-100, CC12M subsets, and a custom CelebA setup introduces distributional differences that likely affect results, particularly for C2 and C3. I believe this makes direct comparison to the original findings more difficult.

- **Sensitivity to hyperparameters and thresholds is underexplored.**
  The paper would benefit from a more systematic analysis of how sensitive the results are to choices such as similarity thresholds and sparsity level, especially when the base model and the datasets are different from the original implementation. We have to rule out that the issues in C2 and C3 are not hyperparameter related.

- **Evaluation of monosemanticity remains partly subjective.**
  Manual inspection plays a significant role. While the centroid-based method is promising, it is not fully integrated into the main evaluation, which weakens the overall analysis.

- **Partial coverage of the original experimental space.**
  Restricting experiments to a single weighting scheme and expansion factor limits the generality of the conclusions.

## Not a weaknesses but encourage further exploration

- **Model vs. data effects are not clearly disentangled.**
  One of the more interesting observations is that the results, particularly for monosemanticity, appear sensitive to dataset composition. In my view, the current evaluation does not clearly separate whether these effects arise from the model itself or from the semantic structure already present in the data (due to smaller dataset) and CLIP embeddings.

**Additional Comments:**

Overall, I think this is a solid and well-executed reproduction study. It successfully validates the main qualitative claims of the original paper while also identifying several important inconsistencies.

While the dataset deviations limit the strength of the reproducibility claims, the paper provides meaningful insights and would be a valuable contribution to the reproducibility track.

**Audience:**

Yes

**Audience Explanation:**

Understanding limitations in self-supervised models and their extensions is of high interest in the community since these limitations are usually underexplored in the main papers

**Claims And Evidence:**

Yes

**Claims Explanation:**

The claims are largely supported by clear and convincing evidence, with consistent reproduction of the main qualitative trends.

**Requested Changes:**

## Suggestions for Improvement

- **Systematically analyse sensitivity.**
  It would be valuable to vary thresholds, k-values, and dataset composition in a controlled way to assess robustness.

- **Strengthen evaluation of monosemanticity.**
  Integrating the centroid-based method into the main evaluation and comparing it directly to threshold-based approaches would improve the analysis.

- **Disentangle model-induced vs. data-induced structure.**
  A particularly useful experiment would be to evaluate the method under controlled reductions in semantic diversity (e.g., restricting to fine-grained subsets or artificially collapsing embedding structure). This would help clarify whether properties such as monosemanticity arise from the model itself or from the underlying data.

---

### Review · Reviewer_bsP9 · 2026-04-27

**Summary Of Contributions:**

This paper is a reproducibility study of Interpreting CLIP with Hierarchical Sparse Autoencoders. The authors focus on four central claims of the original work: that Matryoshka Sparse Autoencoders (MSAEs) improve the sparsity–fidelity trade-off, learn progressive representations, discover monosemantic interpretable concepts, and enable concept-based interventions in CLIP space. The study retrains SAE variants using the original codebase, evaluates MSAE against ReLU and TopK SAE baselines, and reports reconstruction, progressive recovery, concept-discovery, and intervention results. The paper further adds two extensions: a centroid-based method for assessing monosemanticity and an additional qualitative, nearest-neighbor analysis of progressive recovery.


**Strength:**

The main strength of the paper is that it performs a substantive reproduction rather than only reusing reported numbers. It identifies concrete claims, retrains models, documents deviations from the original setting, and reports both agreements and discrepancies. In particular, the paper is useful in showing that the sparsity–fidelity trend is broadly reproducible, while some concept-discovery and neighborhood-preservation results appear sensitive to implementation and evaluation choices.

**Weakness:**

The main weakness is that the paper sometimes states its conclusions more strongly than the evidence supports. Several experiments are performed under downscaled or modified settings, including ImageNet-100 rather than ImageNet-1K, a CC12M subset rather than the original CC3M setup, modified MSAE nesting settings, and an empirically changed cosine threshold for concept discovery. These choices are reasonable for a reproduction study, but they make the strongest conclusion that "the original results are reproduced", which is too broad without further qualification. The C3 and C4 evidence is also partly qualitative and manually validated, so the paper should more carefully separate "reproduced", "mostly reproduced", and "not fully reproduced" findings.

**Audience:**

Yes

**Audience Explanation:**

This paper should be of interest to a subset of the TMLR audience working on vision-language models, sparse autoencoders, interpretability, and reproducibility. The paper provides actionable information about which parts of the original MSAE claims are robust under a downscaled reproduction setting and which parts are sensitive.

**Broader Impact Concerns:**

The paper analyzes concept neurons in CLIP representations and includes experiments involving gender classification and perturbation of gender-associated concepts such as beard, blonde, glasses, suit, and lipstick. This is not necessarily harmful, and in fact can help expose biases in learned representations. However, because the work touches on demographic attributes and stereotype-sensitive interventions, the paper should include a brief broader-impact discussion. This discussion should clarify that the gender experiment is used as a bias-analysis probe, acknowledge the limitations of binary gender classification on CelebA, and caution against using such interventions for demographic inference or profiling.

**Claims And Evidence:**

Yes

**Claims Explanation:**

Overall, I would answer Yes to the evidence criterion, provided the authors revise the wording of the main conclusions. The paper should avoid implying that all original claims are fully reproduced in the abstract. A more accurate conclusion would be that the main sparsity–fidelity and intervention trends are broadly reproduced, while neighborhood preservation and monosemantic concept discovery show sensitivity to dataset choice, preprocessing, thresholding, and validation protocol.

In particular:

For C2, the evidence is mixed. The reconstruction metrics and qualitative nearest-neighbor results support progressive recovery as k increases, but the reported CKNNA behavior differs from the original paper: TopK models, including MSAE, have lower CKNNA than ReLU models in this reproduction. Since this is opposite to the original trend, the claim that C2 is reproduced should be weakened or split into subclaims: reconstruction recovery is reproduced, while neighborhood-preservation behavior is not fully reproduced.

For C3, the paper provides useful evidence, but the conclusion should be more cautious. The authors change the cosine threshold from 0.42 to 0.28 due to a reported preprocessing issue and manually inspect the resulting concepts. They find that only 24 of the 81 detected MSAE concepts are valid in their ImageNet-100 setting, and they also note ambiguity in the definition of monosemanticity. This is valuable evidence, but it supports a claim about sensitivity and partial reproducibility rather than a clean reproduction of monosemantic concept discovery.

For C4, the intervention results are directionally convincing: perturbing gender-associated concepts changes CelebA classifier outputs, and nearest-neighbor examples show interpretable changes such as large-to-tiny and color, quantity changes. However, these results remain mostly qualitative and would benefit from more systematic reporting over multiple images, concepts, perturbation magnitudes, and random seeds.

**Requested Changes:**

**Major:**

1. Calibrate the main reproducibility claim. The abstract and conclusion currently suggest that the original results are broadly reproducible. This should be made more precise. The evidence supports "C1 reproduced", "C2 partially/mixed", "C3 partially/mixed and threshold-sensitive", and "C4 qualitatively reproduced". Please revise the abstract, discussion, and conclusion accordingly.

2. Separate claim-level conclusions more clearly. The paper should provide a compact table summarizing each original claim, the reproduction setting, the main evidence, whether the claim is reproduced/partially reproduced/not reproduced, and the main caveat. This would substantially improve the clarity of the contribution.

3. Clarify the effect of experimental deviations from the original setup. The reproduction uses ImageNet-100 rather than ImageNet-1K, a CC12M subset rather than the original CC3M setup, only expansion factor 8, and only the RW MSAE variant in the main experiments. These are reasonable constraints, but the paper should explicitly discuss which claims are most affected by each deviation. In particular, ImageNet-100 being animal-heavy appears to directly affect concept-discovery validation.

4. Strengthen or narrow C3. The concept-discovery claim is the least cleanly supported. The threshold change from 0.42 to 0.28 should be explained more rigorously, and the manual validation protocol should be described in more detail. At a minimum, the authors should report inter-annotator agreement or a clear rubric for deciding whether a concept is valid or monosemantic. If this is not feasible, the claim should be narrowed to something like "we recover some coherent concept neurons, but automatic concept counts are sensitive to threshold and dataset choice".

5. Clarify the CKNNA discrepancy in C2. Since the CKNNA trend differs from the original paper, the authors should avoid saying that C2 is reproduced without qualification. They should discuss possible causes, such as dataset choice, activation path, implementation differences, or metric sensitivity. If possible, an additional diagnostic experiment would be helpful, but at least the interpretation should be made more cautious.

6. Make C4 more systematic or narrow its claim. The intervention experiments are interesting but mostly qualitative. The authors should either add quantitative summaries over multiple concepts/images/perturbation strengths or explicitly state that the evidence supports qualitative plausibility rather than a broad systematic intervention claim.

**Others:**

1. Several sentences contain grammatical issues or imprecise wording. Please improve the writing and proofreading.
2. For reconstruction metrics, concept counts, and intervention results, reporting variability across seeds or subsets would improve confidence in the findings.
3. As this is a reproducibility study, it would be useful to state exactly what code, configs, checkpoints, and data-processing scripts will be released.

---

### Decision · Action_Editor_ozTB · 2026-06-18

**Recommendation:** Reject

**Additional Comments:**

The authors did not submit any revision/rebuttal after review, meaning that reviewers' comments remain unaddressed. The next version of this manuscript should consider reviewers suggestions and incorporate them to improve the paper.

Reviewers also pointed out that the reproducibility study is conducted using the paper's original codebase. Therefore the next version of this manuscript should comment on the reliability of this codebase (e.g., whether the codebase contains bugs that may lead to incorrect results in the original paper), as well as whether the general idea of the original paper can be reproduced beyond its own codebase.

**Audience:**

Yes

**Audience Explanation:**

The manuscript would be interesting to researchers in multi-modal models and explainable AI.

**Claims And Evidence:**

No

**Claims Explanation:**

This paper is a reproducibility study of the paper "Interpreting CLIP with Hierarchical Sparse Autoencoders".

Reviewers overall welcomed the reproducibility study and commended on the paper in its substantial efforts. However, they are concerned regarding the evidences to support the claims within the reproducibility study.

- Reviewer bsP9 considered the evidence for claim C2 mixed, in particular they had doubts with the reported CKNNA behaviour. Reviewer fcug also raised doubts regarding the "reproduced" conclusions for claims C1 & C2.
- Reviewers bsP9 and Smky pointed out that the evidence for claim C4 is largely qualitative, and they suggested a tone-down of the claim, or improved experimental settings to solidify the evidence.

**Resubmission Of Major Revision:**

The authors may consider submitting a major revision at a later time.